

**1** **An observational constraint on stomatal function in forests:**

**2** **evaluating coupled carbon and water vapor exchange with**

**3** **carbon isotopes in the Community Land Model (CLM 4.5)**

**4** **Brett Raczka[1], Henrique F. Duarte[2], Charles D. Koven[3], Daniel Ricciuto[4], Peter E.**

**5** **Thornton[4], John C. Lin[2], David R. Bowling[1]**

**6** [1]{Dept. of Biology, University of Utah, Salt Lake City, Utah}

**7** [2]{Dept. of Atmospheric Sciences, University of Utah, Salt Lake City, Utah}

**8** [3]{Lawrence Berkeley National Laboratory, Berkeley, California}

**9** [4]{Oak Ridge National Laboratory, Oak Ridge, Tennessee}

**11** Correspondence to: B. Raczka (brett.raczka@utah.edu)

**12** **Abstract**

**13** Land surface models are useful tools to quantify contemporary and future climate impact on

**14** terrestrial carbon cycle processes, provided they can be appropriately constrained and tested

**15** with observations. Stable carbon isotopes of $CO_2$ offer the potential to improve model

**16** representation of the coupled carbon and water cycles because they are strongly influenced by

**17** stomatal function. Recently, a representation of stable carbon isotope discrimination was

**18** incorporated into the Community Land Model component of the Community Earth System

**19** Model. Here, we tested the model's capability to simulate whole-forest isotope discrimination

**20** in a subalpine conifer forest at Niwot Ridge, Colorado, USA. We distinguished between

**21** isotopic behavior in response to a decrease of $\delta^{13}C$ within atmospheric $CO_2$ (Suess effect) vs.

**22** photosynthetic discrimination ($\Delta_{canopy}$), by creating a site-customized atmospheric $CO_2$ and

**23** $\delta^{13}C$ of $CO_2$ time series. We implemented a seasonally-varying $V_{cmax}$ model calibration that

**24** best matched site observations of net $CO_2$ carbon exchange, latent heat exchange and biomass.

**25** The model accurately simulated observed $\delta^{13}C$ of needle and stem tissue, but underestimated

**26** the $\delta^{13}C$ of bulk soil carbon by 1-2 ‰. The model overestimated the multi-year (2006-2012)

**27** average $\Delta_{canopy}$ relative to prior data-based estimates by 5-6 ‰. The amplitude of the average

**28** seasonal cycle of $\Delta_{canopy}$ (i.e. higher in spring/fall as compared to summer) was correctly

**29** modeled but only with an alternative nitrogen limitation formulation for the model. The model



attributed most of the seasonal variation in discrimination to the net assimilation rate ($A_n$),
whereas inter-annual variation in simulated $\Delta_{canopy}$ during the summer months was driven by
stomatal response to vapor pressure deficit. Soil moisture did not influence modeled $\Delta_{canopy}$.
The model simulated a 10% increase in both photosynthetic discrimination and water use
efficiency (WUE) since 1850 as a result of $CO_2$ fertilization, forced by constant climate
conditions. This increasing trend in discrimination is counter to well-established relationships
between discrimination and WUE. The isotope observations used here to constrain CLM
suggest 1) the model overestimated stomatal conductance and 2) the default CLM approach to
representing nitrogen limitation (post-photosynthetic limitation) was not capable of
reproducing observed trends in discrimination. These findings demonstrate that isotope
observations can provide important information related to stomatal function driven by
environmental stress from VPD and nitrogen limitation.

## 1   Introduction

16        The net uptake of carbon by the terrestrial biosphere currently mitigates the rate of

atmospheric $CO_2$ rise and thus the rate of climate change. Approximately 25% of
anthropogenic $CO_2$ emissions are absorbed by the global land surface (Le Quéré et al., 2015),
but it is unclear how projected changes in temperature and precipitation will influence the future
of this land carbon sink (Arora et al., 2013; Friedlingstein et al., 2006). A major source of
uncertainty in climate model projections results from the disagreement in projected strength of
the land carbon sink (Arora et al., 2013). Thus, it is critical to reduce this uncertainty to improve
climate predictions, and to better inform mitigation strategies (Yohe et al., 2007).

24        An effective approach to reduce uncertainties in terrestrial carbon models is to constrain

a broad range of processes using distinct and complementary observations. Traditionally,
terrestrial carbon models have relied primarily upon observations of land-surface fluxes of
carbon, water and energy derived from eddy-covariance flux towers to calibrate model
parameters and evaluate model skill. Flux measurements best constrain processes that occur at
diurnal and seasonal time scales (Braswell et al., 2005; Ricciuto et al., 2008). Traditional
ecological metrics of carbon pools (e.g. leaf area index, biomass) are also commonly used to
provide independent and complementary constraints upon ecosystem processes at longer time
scales (Ricciuto et al., 2011; Richardson et al., 2010). However, neither flux nor carbon pool



observations provide suitable constraints for the model formulation of plant stomatal function
and the related link between the carbon and water cycles.
Stable carbon isotopes of $CO_2$ are influenced by stomatal activity in C3 plants (e.g.
evergreen trees, deciduous trees), and thus provide a valuable but under-utilized constraint on
terrestrial carbon models.  Plants assimilate more of the lighter of the two major isotopes of
atmospheric carbon ($^{12}C$ vs. $^{13}C$).   This preference, termed photosynthetic discrimination
($\Delta_{canopy}$), is primarily a function of two processes, $CO_2$ diffusion rate through the leaf boundary
layer and into the stomata, and the carboxylation of $CO_2$.  The magnitude of $\Delta_{canopy}$ is controlled
by $CO_2$ supply (atmospheric $CO_2$ concentration, stomatal conductance) and demand
(photosynthetic rate; Flanagan et al., 2012).  In general, environmental conditions favorable to
plant productivity result in higher $\Delta_{canopy}$ during carbon assimilation compared to unfavorable
conditions.  Plants respond to unfavorable conditions by closing their stomata and reducing the
stomatal conductance which reduces $\Delta_{canopy.}$  Most relevant here, $\Delta_{canopy}$ responds to
atmospheric moisture deficit (Andrews et al., 2012; Wingate et al., 2010), soil water content
(McDowell et al., 2010), precipitation (Roden and Ehleringer, 2007) and nutrient availability.
After carbon is assimilated, additional post-photosynthetic isotopic changes occur (Bowling et
al., 2008; Brüggemann et al., 2011), but these impose a small influence on land-atmosphere
isotopic exchange relative to photosynthetic discrimination.
The Niwot Ridge Ameriflux site, located in a sub-alpine conifer forest in the Rocky
Mountains of Colorado, U.S.A., has a long legacy of yielding valuable datasets to test carbon
and water functionality of land surface models using stable isotopes.  Niwot Ridge has a 17-
year record of eddy covariance fluxes of carbon, water, and energy, as well as environmental
data (Hu et al., 2010; Monson et al., 2002) and a 10-year record of $\delta^{13}C$ of $CO_2$ in forest air
(Schaeffer et al., 2008).   From a carbon balance perspective, Niwot Ridge is representative of
subalpine forests in Western North America that, in general, act as a carbon sink to the
atmosphere (Desai et al., 2011).  Western forests, make up a significant portion of the carbon
sink in the United States (Schimel et al., 2002), yet this carbon sink is projected to weaken with
projected changes in temperature and precipitation (Boisvenue and Running, 2010).
The Community Land Model (CLM), the land sub-component of the Community Earth
System Model (CESM) has a comprehensive representation of biogeochemical cycling (Oleson
et al., 2013) that can be applied across a range of temporal  (hours to centuries) and spatial
scales (site to global).   A mechanistic representation of photosynthetic discrimination based





upon diffusion and enzymatic fractionation (Farquhar et al., 1989) was included in the latest
release of CLM 4.5 (Oleson et al., 2013). An early version of CLM simulated carbon (but not
carbon isotope) dynamics at Niwot Ridge with reasonable skill (Thornton et al., 2002). To date,
we are not aware of any CLM-based studies that have used $CO_2$ isotopes at natural abundance
to quantify the accuracy of the photosynthetic discrimination sub-model, or to evaluate the
utility of $CO_2$ isotopes to constrain carbon and water cycle coupling.
Here, we evaluate the performance of the $^{13}C/^{12}C$ isotope discrimination sub-model
within CLM 4.5 against a range of isotopic observations at Niwot Ridge, to examine what new
insights an isotope-enabled model can bring upon ecosystem function. Specifically, we test
whether CLM simulates the expected isotopic response to environmental drivers of $CO_2$
fertilization, soil moisture and atmospheric vapor pressure deficit (VPD). A previous analysis
at Niwot Ridge showed a seasonal correlation between vapor pressure deficit (VPD) and
photosynthetic discrimination (Bowling et al., 2014) which may suggest that leaf stomata are
responding to changes in VPD, and influencing discrimination. We use CLM to test whether
VPD is the primary environmental driver of isotopic discrimination, as compared to soil
moisture and net assimilation rate. Next we determine whether including site-specific $\delta^{13}C$ of
atmospheric $CO_2$ within the model simulation combined with simulated long term (multi-
decadal to century) photosynthetic discrimination and simulated carbon pool turnover,
accurately reproduces the measured $\delta^{13}C$ in leaf tissue, roots and soil carbon. We then use CLM
to determine if the increase in atmospheric $CO_2$ since 1850 has led to an increase in WUE, and
whether net assimilation or stomatal conductance is the primary driver of such a change.
Finally, we ask what distinct insights site level isotope observations bring in terms of both
model parameterization (i.e. stomatal conductance) and model structure as compared to the
traditional observations (e.g. carbon fluxes, biomass).

## 2  Methods

We focus the description of CLM 4.5 (Section 2.1) upon photosynthesis, and its linkage
to nitrogen, soil moisture and stomatal conductance (Section 2.1.1). Next we describe the
model representation of carbon isotope discrimination by photosynthesis (Section 2.1.2).
Because preliminary simulations demonstrated that model results were strongly influenced by
nitrogen limitation, we used three separate nitrogen formulations (described in Section 2.1.2)
to better diagnose model performance. Next, to provide context for subsequent descriptions of





site-specific model adjustments we describe the field site, Niwot Ridge, including the site level
observations (Section 2.2) used to constrain model behavior and test model skill.

3        Patterns in plant growth and $\delta^{13}C$ of biomass are strongly influenced by atmospheric $CO_2$

and $\delta^{13}C$ of atmospheric $CO_2$  ($\delta_{atm}$). Therefore we designed a site-specific synthetic
atmospheric $CO_2$ product (Section 2.3.1) and $\delta_{atm}$ product (Section 2.3.2) for these simulations.
The model setup and initialization procedure, intended to bring the system into steady state, is
described in Section (2.3.3).   This is followed by an explanation of the model calibration
procedure that provided a realistic simulation of carbon and water fluxes (Section 2.4).
**2.1  Community Land Model, Version 4.5**

10        We used the Community Land Model, CLM 4.5 (Oleson et al., 2013), which is the land

component of the Community Earth System Model (CESM) version 1.2
(https://www2.cesm.ucar.edu/models/current).  Details regarding the Community Land Model
can be found in (Mao et al., 2016; Oleson et al., 2013).   Here, we emphasize the mechanistic
formulation that controls photosynthetic discrimation ($\Delta_{canopy}$) and factors that influence
$\Delta_{canopy}$ including photosynthesis, stomatal conductance, water stress and nitrogen limitation. A
list of symbols is provided in Table (1).

### 2.1.1  Net Photosynthetic Assimilation

18        The net carbon assimilation of photosynthesis, $A_n$ is based on Farquhar et al., (1980) as,

$$A_n = \min(A_c, A_j, A_p) - Resp_d, \tag{1}$$
where $A_c$, $A_j$ and $A_p$ are the enzyme (Rubisco)-limited, light-limited, and product-limited  rates
of carboxylation respectively, and $Resp_d$ the leaf-level dark respiration.  The enzyme limited
rate is defined as
$$A_c = \frac{V_{cmax}(c_i - \Gamma_*)}{c_i + K_c(1 + \frac{o_i}{K_o})}, \tag{2}$$
where $c_i$ is the internal leaf partial pressure of $CO_2$, $o_i = 0.209\, P_{atm}$, where $P_{atm}$ is atmospheric
pressure, and $K_c$, $K_o$ and $\Gamma_*$ are constants.  The maximum rate of carboxylation at 25°C, $V_{cmax25}$,
is defined as
$$V_{cmax25} = N_a\ F_{LNR}\ F_{NR}\ a_{R25}, \tag{3}$$



where $N_a$ is the nitrogen concentration per leaf area, $F_{LNR}$ the fraction of leaf nitrogen within
the Rubisco enzyme, $F_{NR}$ the ratio of total Rubisco molecular mass to nitrogen mass within
Rubisco, and $a_{R25}$ is the specific activity of Rubisco at 25°C. The $V_{cmax25}$ is adjusted for leaf
temperature to provide $V_{cmax}$ in Eq. 2, used in the final photosynthetic calculation.
The carbon and water balance are linked through $c_i$ by the stomatal conductance, $g_s$,
following the Ball-Berry model as defined by Collatz et al., (1991),
$$g_s = m \frac{A_n}{c_s/P_{atm}} h_s + b\beta_t,$$  (4)
where $m$ is the stomatal slope, $c_s$ the partial pressure of CO$_2$ at the leaf surface and $b$ the
minimum stomatal conductance when the leaf stomata are closed. The variable $h_s = e_l/e_s$ is
the leaf surface specific humidity with $e_l$ the vapor pressure at the leaf surface and $e_s$ the
saturation vapor pressure inside the leaf. The variable $\beta_t$ represents the level of soil moisture
availability, which influences stomatal conductance directly, but also indirectly through $A_n$ by
multiplying $V_{cmax}$ by $\beta_t$ (Sellers et al., 1996). CLM calculates $\beta_t$ as a factor (0-1, high to low
stress) by combining soil moisture, the rooting depth profile, and a plant-dependent response to
soil water stress as
$$\beta_t = \sum_i w_i r_i,$$  (5)
where $w_i$ is a plant wilting factor for soil layer $i$ and $r_i$ is the fraction of roots in layer $i$. The
plant wilting factor is scaled according to soil moisture and water potential, depending on plant
functional type (PFT). Soil moisture is predicted based upon prescribed precipitation and
vertical soil moisture dynamics (Zeng and Decker, 2009). The root fraction in each soil layer
depends upon a vertical exponential profile controlled by PFT dependent root distribution
parameters adopted from Zeng (2001).
The version of CLM used here has a 2-layer (shaded, sunlit) representation of the tree
canopy. Photosynthesis and stomatal conductance are calculated separately for the shaded and
sunlit portion and the total canopy photosynthesis is the potential gross primary productivity
(GPP), $CF_{GPPpot}$. The total carbon available for new growth allocation ($CF_{avail\_alloc}$) is
defined as
$$CF_{avail\_alloc} = CF_{GPPpot} - CF_{GPP,mr} - CF_{GPP,xs},$$  (6)
where $CF_{GPP,mr}$ is the carbon costs for maintenance respiration and $CF_{GPP,xs}$ is the carbon
allocated to a pool responsible for meeting maintenance respiration demand during periods with





low or zero photosynthesis. In contrast, $CF_{alloc}$, is the actual carbon allocated to growth
calculated from the available nitrogen and fixed C:N ratios for new growth (e.g. stem, roots,
leaves). The downregulation of photosynthesis from nitrogen limitation, $f_{dreg,}$ is given by

$$f_{dreg} = \frac{CF_{avail\_alloc} - CF_{alloc}}{CF_{GPPpot}} . \tag{7}$$

### 2.1.2 Photosynthetic Carbon Isotope Discrimination

The canopy-level fractionation factor $\alpha_{psn}$ is defined as the ratio of $^{13}C/^{12}C$ within
atmospheric $CO_2$ ($R_a$) and the products of photosynthesis ($R_{GPP}$) as $\alpha_{psn} = \frac{R_a}{R_{GPP}}$. The
preference of C3 vegetation to assimilate the lighter $CO_2$ molecule during photosynthesis is
simulated in CLM with two steps: diffusion of $CO_2$ across the leaf boundary layer and into the
stomata, followed by enzymatic fixation to give the leaf-level fractionation factor:

$$\alpha_{psn} = 1 + \frac{4.4 + 22.6 \frac{c_i^*}{c_a}}{1000} . \tag{8}$$

where $c_i^*$ and $c_a$ are the intracellular and atmospheric $CO_2$ partial pressure respectively. The
variable $c_i^*$ is marked with an asterisk to indicate the inclusion of nitrogen downregulation as
defined as,

$$c_i^* = c_a - A_n (1 - f_{dreg}) P_{atm} \frac{(1.4g_s) + (1.6g_b)}{g_b g_s} \tag{9}$$

where $g_b$ is the leaf boundary layer conductance. The inclusion of the nitrogen downregulation
factor $f_{dreg}$ in the above expression reflects the two-stage process in which the potential
photosynthesis and the actual photosynthesis are calculated within CLM and prevents a
mismatch between the actual photosynthesis and the intracellular $CO_2$.
The sensitivity of preliminary model results to nitrogen limitation led us to test three
distinct discrimination formulations (Table 2). The *limited nitrogen* formulation, was based on
the default version of CLM 4.5 and included both nitrogen limitation and the nitrogen
downregulation factor within the calculation of $c_i^*$ as given in equation (9). In the second,
*unlimited nitrogen* formulation, we allowed vegetation to have unlimited access to nitrogen
($CF_{GPPpot} = CF_{GPP}$, $f_{dreg}=0$). Finally, in the *no downregulation discrimination* formulation,
we included nitrogen limitation, but removed the downregulation factor $f_{dreg}$ from equation

27 (9).





In the *unlimited nitrogen* formulation, we use a different modifier on V$_{cmax25}$ (described
in section 2.4 and Fig. S1, S2) in the calibrated runs to give similar carbon flux, water flux and
biomass as in the other two formulations, such that all three formulations have fluxes and
biomass that are similar to what is observed at the site, and which presumably reflect nitrogen
limitation.  Thus the distinction between these three formulations can be viewed entirely of
when nitrogen limitation is imposed in relation to photosynthesis: (1) *after photosynthesis* via
a downregulation between potential and actual GPP (equation 7) that feeds back on the $c_i/c_a$
used for isotopic discrimination but not on the stomatal conductance in the *limited nitrogen*
formulation; (2) *before photosynthesis* via V$_{cmax,}$ which limits photosynthetic capacity affecting
both $c_i/c_a$ and stomatal conductance in the *unlimited nitrogen* formulation; and (3) *after*
*photosynthesis with no effect on either* the $c_i/c_a$ for isotopic discrimination or the stomatal
conductance in the *no downregulation discrimination* formulation.

13       Carbon isotope ratios are expressed by standard delta notation,

$$\delta^{13}C_x = \left( \frac{R_x}{R_{VPDB}} - 1 \right) \times 1000, \tag{10}$$
where $R_x$ is the isotopic ratio of the sample of interest, and $R_{VPDB}$ is the isotopic ratio of the
Vienna Pee Dee Belemnite standard.  The delta notation is dimensionless but expressed in parts
per thousand (‰) where a positive (negative) value refers to a sample that is enriched (depleted)
in $^{13}C/^{12}C$ relative to the standard.  Because this is the only carbon isotope ratio we are
concerned with in this paper, the '13' superscript is omitted for brevity in subsequent definitions
using the delta notation.  The canopy-integrated photosynthetic discrimination, $\Delta_{canopy}$, is
defined as the difference between the $\delta^{13}C$ of the atmospheric and assimilated carbon,
$$\Delta_{canopy} = \delta_{atm} - \delta_{GPP.} \tag{11}$$
The difference between $\delta^{13}C$ of the total ecosystem respiration (ER) and GPP fluxes, called the
isotope disequilibrium (Bowling et al., 2014), is defined as,
$$disequilibrium = \delta_{ER} - \delta_{GPP.} \tag{12}$$

26       The ecosystem-level water use efficiency (WUE) is defined as carbon assimilated ($GPP$)

per unit water transpired ($E_T$) per unit land surface area,
$$WUE = \frac{GPP}{E_T}. \tag{13}$$
The intrinsic water use efficiency ($iWUE$) from leaf-level physiological ecology is defined as,



$iWUE = \frac{A}{g_s}$ ,                                                   (14)
where $A$ is the carbon assimilated per unit leaf area and $g_s$ is the stomatal conductance. CLM
calculates $g_s$ (Equation 4) for shaded and sunlit portions of the canopy separately, therefore an
overall conductance was calculated by weighting the conductance by sunlit and shaded leaf
areas and is used in this manuscript.
**2.2   Niwot Ridge and site-level observations**

7        Site-level observations and modeling were focused on the Niwot Ridge Ameriflux

tower, a sub-alpine conifer forest located in the Rocky Mountains of Colorado, U.S.A. The
forest is approximately 110 years old and consists of lodgepole pine, Engelmann spruce, and
subalpine fir. The site is located at an elevation of 3050 m above sea level, with mean annual
temperature of 1.5°C and precipitation of 800 mm, in which approximately 60% is snow.
More site details are available elsewhere (Hu et al., 2010; Monson et al., 2002). Flux and
meteorological data were obtained from the Ameriflux archive (http://ameriflux.lbl.gov/).
Net carbon exchange (NEE) observations from the flux towers were partitioned into component
fluxes of GPP and ER according to methods described by Reichstein et al., (2005) and Lasslop
et al., (2010) using an online tool provided by the Max Planck Institute ([http://www.bgc-](http://www.bgc-jena.mpg.de/~MDIwork/eddyproc/)
[jena.mpg.de/~MDIwork/eddyproc/](http://www.bgc-jena.mpg.de/~MDIwork/eddyproc/)). Seasonal patterns in $\delta_{GPP}$ and $\delta_{ER}$ were derived from
measurements as described by (Bowling et al., 2014). Observations of $\delta^{13}C$ of biomass
(Schaeffer et al., 2008) and carbon stocks (Bradford et al., 2008; Scott-Denton et al., 2003)
were compared to model simulations. Schaeffer et al., (2008) reported soil, leaf and root
observations specific to each conifer species, however, the observed mean and standard error
for all species were used for comparison because CLM treated all conifer species as a single
PFT.
**2.3   Atmospheric CO₂, isotope forcing and initial vegetation state**
2.3.1   Site-specific atmospheric $CO_2$ concentration time series

26        Global average atmospheric $CO_2$ concentrations increased roughly 40% from 1850 to

2013 (from 280 to 395 ppm). The standard version of CLM 4.5 includes an annually and
globally averaged time series of this $CO_2$ increase, however, this does not capture the observed
seasonal variation of ~10 ppm at Niwot Ridge (Trolier et al., 1996). Therefore we created a



site-specific atmospheric $CO_2$ time series (Figure 1) to provide a seasonally realistic atmosphere
at Niwot Ridge.   Observations were used to create the synthetic product from 1968-2013 by
binning flask observations into 20 evenly spaced points each year.  These flask observations
were taken weekly from Niwot Ridge (Dlugokencky et al., 2015).  Prior to 1968, a polynomial
fit of the annualized CLM product was created and then adjusted by 1.5 ppm to account for the
average difference between the CLM product and the Niwot Ridge observations during those
years.  Next, the average multi-year seasonal cycle based on the de-trended flask data after 1968
was added to every year of this annualized polynomial before 1968.  Finally, the synthetic
atmospheric $CO_2$ time series (pre 1968) was populated with 20 evenly spaced points in time
each year.

## 2.3.2  Customized $\delta^{13}C$ atmospheric $CO_2$ time series

13         As atmospheric $CO_2$ has increased, the $\delta^{13}C$ of atmospheric $CO_2$ ($\delta_{atm}$) has become

more depleted (Francey et al., 1999), and this change has occurred at Niwot Ridge at -0.25 ‰
per decade (Bowling et al., 2014).  The $\delta_{atm}$ also varies seasonally, and depends on latitude
(Trolier et al., 1996).  However, CLM 4.5 as released assigned a constant $\delta^{13}C$ of -6 ‰.  We
therefore created a synthetic time series of $\delta_{atm}$ from 1850-2013 (Figure 1).   From 1990-2013
this was based upon the flask observations (White et al., 2015) as described in Section 2.3.1.  A
similar approach to the atmospheric $CO_2$ synthetic time series (Section 2.3.1) was applied here
to create the synthetic $\delta_{atm}$.  After 1990 the flask data were binned into 20 evenly spaced points
each year.  Prior to 1990 the inter-annual variation was based upon a polynomial fit to ice core
data from Law Dome (Francey et al., 1999; see also Rubino et al., 2013).  The polynomial was
adjusted by 0.20 ‰ to account for the inter-hemispheric difference identified during the
common years (1990-1996) between the ice core and flask data.  Next the average seasonal
cycle (1990-2013) of $\delta_{atm}$ was added to the adjusted polynomial prior to 1990. The synthetic
time series was populated from 1850-1989 with 20 evenly spaced points each year based upon
the adjusted polynomial with seasonal cycle included.  As released, CLM 4.5 was not
compatible with time varying $\delta_{atm}$,  therefore we modified the source code by following the
model procedure for reading in time-varying $^{14}C$.   The modified code was designed to
temporally interpolate the $\delta_{atm}$ time series for each time step of the model.   This interpolated



value was then passed into the photosynthetic discrimination calculation to represent the time-
varying $\delta_{atm}$.

### 2.3.3  Model Initialization

We performed an initialization to transition the model from near bare-ground conditions
to present day carbon stocks and LAI that allowed for proper evaluation of isotopic
performance.  This was implemented in 4 stages: 1) accelerated decomposition (1000 model
years) 2) normal decomposition (1000 model years) 3) parameter calibration (1000 model
years) and 4) transient simulation period (1850-2013).  The first two stages were pre-set options
within CLM with the first stage used to accelerate the equilibration of the soil carbon pools,
which require a long period to reach steady state (Thornton and Rosenbloom, 2005).  The
parameter calibration stage was not a pre-set option but designed specifically for our analysis.
For this we introduced a seasonally varying $V_{cmax}$ that scaled the simulated GPP and ecosystem
respiration fluxes to present day observations (Section 2.4).  In the transient phase, we
introduced time-varying atmospheric conditions from 1850-2013 including nitrogen deposition
(CLM provided), atmospheric $CO_2$, and $\delta_{atm}$ (site-specific as described above).  Environmental
conditions of temperature, precipitation, relative humidity, radiation, and wind speed were
taken from the Niwot Ridge flux tower observations from 1998-2013 and then cycled
continuously for the entirety of the initialization process.   We used a scripting framework
(PTCLM) that automated much of the workflow required to implement several of these stages
in a site level simulation (Mao et al., 2016; Oleson et al., 2013).

### 2.4  Specific model details and model calibration

We used PTCLM (e.g. Mao et al., 2016) to create site specific weather conditions and
initial conditions for CLM 4.5. This version of CLM included a fully prognostic representation
of carbon and nitrogen within its vegetation, litter and soil biogeochemistry.  We used the
Century model representation for soil (3 litter and 3 soil organic matter pools) with 15 vertically
resolved soil layers.  Nitrification and prognostic fire were turned off.   Our initial simulations
used prognostic fire, but we found that simulated fire was overactive leading to low simulated
biomass compared to observations. Although Niwot Ridge has been subject to disturbance from



fire and harvest in the past, ultimately our final simulations did not include either fire or harvest
disturbance because the last disturbance occurred over 100 years ago (early 20th century
logging; Monson et al., 2005).

4        Ecosystem parameter values (Table 3) used here were based upon the temperate

evergreen needleleaf plant functional type (PFT) within CLM.  These values were based upon
observations reported by White et al., (2000) intended for a wide range of temperate evergreen
forests, and by Thornton et al., (2002) for Niwot Ridge.   For this analysis two site-specific
parameter changes were made.  First, the e-folding soil decomposition parameter was increased
from 5 to 20 meters. This parameter is a length-scale for attenuation of decomposition rate for
the resolved soil depth from 0 to 5 meters where an increased value effectively increases
decomposition at depth, thus reducing total soil carbon and more closely matching
observations.  Second, we performed an empirical photosynthesis scaling (equation 15, below)
that reduced the simulated photosynthetic flux, as guided by eddy covariance observations.
Consequently, all downstream carbon pools and fluxes including ecosystem respiration,
aboveground biomass, and leaf area index which provided a better match to present day
observations.  This approach also removed a systematic overestimation of winter
photosynthesis. The model simulations without the photosynthetic scaling are referred to within
the text and figures as the *uncalibrated* model, whereas  model simulations that include the
photosynthetic scaling are referred to as the *calibrated* model.   The source code was modified
for this scaling approach by reducing $V_{cmax}$ at 25º Celsius,
$V_{cmax25} = N_a \ F_{LNR} \ F_{NR} \ a_{R25} \ f_{df},$                                                  (15)
where $f_{df}$ is the photosynthetic scaling factor,  and all other parameters are identical to equation
(3).  These parameters were constant for the entirety of the simulations except for $f_{df}$, an
empirically derived time dependent parameter ranging from 0-1.  The value was set to zero to
force photosynthesis to zero between November 13th and March 23rd, consistent with flux
tower observations where outside of this range GPP > 0 was never observed.  During the
growing season period (GPP>0) within days of year 83-316, $f_{df}$ was calculated as
$f_{df} = \frac{observed\ GPP(day\ of\ year)}{simulated\ GPP(day\ of\ year)}, \ \ 82 < day\ of\ year < 317$                        (16)
where the *observed GPP* was the  daily average calculated from the partitioned flux tower
observations (Reichstein et al., 2005) from 2006-2013, and the *simulated GPP* was the daily
average of the unscaled value during the same time.  A polynomial was fit to equation (16) that



represented $f_{df}$ for 1) both the *limited nitrogen* and *no downregulation discrimination*
*formulation* and 2) the *unlimited nitrogen* formulation (Figure S1). Note that CLM already
includes a *daylength factor* that also adjusts the magnitude of $V_{cmax}$ according to time of year,
however, that default parameterization alone was not sufficient to match the observations.

## 3    Results & Discussion

8         This section is organized into four parts. First the calibrated model performance is

evaluated against observed bulk carbon pool and bulk carbon flux behavior (Section 3.1.1), and
against the observed $\delta^{13}C$ within carbon pools (Section 3.1.2). Second, the simulated
photosynthetic discrimination is evaluated for multi-decadal trends (Section 3.2.1), magnitude
(Section 3.2.2) and seasonal patterns (Section 3.2.3), including the environmental factors that
were most responsible for driving the seasonal discrimination (Section 3.2.4). Third, we discuss
how isotope observations can be used to guide model development related to nitrogen limitation
(Section 3.3). Finally, we evaluate the capability of the model to reproduce the magnitude and
trends of disequilibrium (Section 3.4).

### 3.1    Calibrated model performance

#### 3.1.1    Fluxes & carbon pools

19         The CLM model was successful at simulating GPP, ER, and latent heat fluxes (Fig. 2),

leaf area index (LAI), and aboveground biomass (Fig. 3), but only following site-specific
calibration. The *uncalibrated* simulation (*limited nitrogen* formulation) overestimated LAI (39
%), aboveground biomass (48%), average peak warm season GPP (15%), and average peak
warm season ER (40%) and overestimated cold-season GPP by 200 gC m$^{-2}$ yr$^{-1}$. The *calibrated*
simulation was much closer to the observations for LAI and aboveground biomass (Figure 3).
The calibrated peak warm and cold season GPP, and warm season ER matched observations.
The simulated latent heat fluxes were relatively insensitive to the calibration. Overall the
simulated latent heat during the warm season overestimated the observations by 10% and
underestimated by 10% during the cold season. Similar improvement was observed after
calibration for the *unlimited nitrogen* run (not shown).





The calibration also eliminated erroneous winter GPP.   In general, terrestrial carbon
models tend to overestimate photosynthesis during cold periods for temperate/boreal conifer
forests (Kolari et al., 2007), including Niwot Ridge (Thornton et al., 2002).  One approach to
correct for this is to include an acclimatization temperature (e.g. Flanagan et al., 2012) that
reduces photosynthetic capacity during the spring and fall.  The CLM 4.5 model includes
functionality to adjust the photosynthetic capacity, including both a temperature acclimatization
and a day length factor that reduces $V_{cmax}$ (Bauerle et al., 2012; Oleson et al., 2013).  However,
this alone was not sufficient to match the observed fluxes.  Although our calibration approach
forced $V_{cmax}$ to zero during the winter, it did not solve the underlying mechanistic shortcoming.
A more fundamental approach should address either cold inhibition (Zarter et al., 2006) of
photosynthesis or root access to soil moisture (Monson et al., 2005) to achieve the
photosynthetic reduction.  Nevertheless, within the confines of our study area, our calibration
approach was sufficient to provide a skillful representation of photosynthesis and provided a
sufficient testbed for evaluating carbon isotope behavior.

### 3.1.2   $\delta^{13}$C of carbon pools

The model performed better simulating $\delta^{13}$C biomass of bulk needle tissue, roots and soil
carbon (Figure 4) for the *unlimited nitrogen* and no *downregulation discrimination* cases as
compared to the *limited nitrogen* case.  When nitrogen limitation was included the model
underestimated $\delta^{13}$C of sunlit needle tissue (1.8 ‰), bulk roots (1.0 ‰), and organic soil carbon
(0.7‰).  All simulations fell within the observed range of $\delta^{13}$C in needles that span from -28.7
‰ (shaded) to -26.7 (sunlit).  This vertical pattern in $\delta^{13}$C of leaves is common (Martinelli et
al., 1998) and results from vertical differences in nitrogen allocation and photosynthetic
capacity.  The model results integrated the entire canopy and ideally should be closer to sun
leaves (as in Figure 4) given that the majority of photosynthesis occurs near the top of the
canopy.
Model simulations of $\delta^{13}$C of living roots were ~1 ‰ more negative as compared to the
structural roots. This range in $\delta^{13}$C results from decreasing $\delta_{atm}$ with time (Suess effect, Figure
1). The living roots had a relatively fast turnover time of carbon within the model, whereas the
structural roots had a slower turnover time and reflected an older (more enriched $\delta_{atm}$)
atmosphere.  The *limited nitrogen* simulation was a poor match to observations relative to the
others (Figure 4, middle panel).





There was an observed vertical gradient in $\delta^{13}C$ of soil carbon (-24.9 to -26 ‰) with more
enriched values at greater depth (Figure 4, right panel).   This vertical gradient is commonly
observed (Ehleringer et al., 2000).  Simulated $\delta^{13}C$ of soil carbon was most consistent with the
organic horizon observations.  There are a wide variety of post-photosynthetic fractionation
processes in the soil system (Bowling et al., 2008; Brüggemann et al., 2011) that are not
considered in the CLM 4.5 model, so the match with observations is perhaps fortuitous.
**3.2  Photosynthetic discrimination**
3.2.1  Decadal changes in photosynthetic discrimination and driving factors
All modeled carbon pools showed steady depletion in $\delta^{13}C$ since 1850 (coinciding with
the start of the transient phase of simulations, Figure 4).  For the *limited nitrogen* run, there was
a decrease in $\delta^{13}C$ of 2.3 ‰ for needles, 2.3 ‰ for living roots, and 0.1 ‰ for soil carbon.  This
occurred because of 1) decreased $\delta_{atm}$ (Suess effect, Figure 1) and 2) increased photosynthetic
discrimination. We quantified the contribution of the Suess effect by performing a control run
with constant $\delta_{atm}$, and kept other factors the same (Figure 5).  Approximately 70% of the
reduction in $\delta^{13}C$ of needles occured due to the Suess effect, and the remaining 30% was caused
by increased photosynthetic discrimination.   This occurred as plants responded to $CO_2$
fertilization as illustrated in Figure (6).  The model indicated that plants responded to increased
atmospheric $CO_2$ (~40% increase) by decreasing stomatal conductance (Equation 4)  by 20%
for the *limited nitrogen* run and 30% for the *unlimited nitrogen* run (Figure 6B) with associated
change in $c_i/c_a$ (Figure 6A).  Other influences upon stomatal conductance were less significant,
including $A_n$ (+ 10% *limited nitrogen*, -10% *unlimited nitrogen*, Figure 6D), soil moisture
availability (2-3%, Figure 6E), and negligible changes in relative humidity (potential climate
change effects are neglected due to methodological cycling of weather data).  This finding that
stomatal conductance responded to atmospheric $CO_2$ is consistent with both tree ring studies
(Saurer et al., 2014) and flux tower measurements (Keenan et al., 2013).
The effect of $CO_2$ fertilization, and associated response of stomatal conductance and net
assimilation led to a multi-decadal increase in $c_i/c_a$ for all model formulations (Figure 6A).  The
$c_i/c_a$ increased from 0.71 to 0.76, 0.67 to 0.71 and 0.66 to 0.68 for the *limited nitrogen*, *unlimited*
*nitrogen* and *no downregulation discrimination* formulations respectively from 1850-2013.  All
simulations therefore suggested an *increase* in photosynthetic discrimination.  This increase in
discrimination falls in between two hypotheses posed by Saurer et al., (2004) regarding stomatal





response to increased $CO_2$: 1) reduction in stomatal conductance causes $c_i$ to proportionally
increase with $c_a$ keeping $c_i/c_a$ constant and 2) minimal stomatal conductance response where $c_i$
increases at the same rate as $c_a$ (constant $c_a - c_i$) causing $c_i/c_a$ to increase. Our simulation
generally agrees with the observed trend in $c_i/c_a$ as estimated from tree ring isotope
measurements from a network of European forests (Frank et al., 2015). When controlled for
trends in climate, Frank et al. (2015) found that $c_i/c_a$ was approximately constant during the last
century. If the Niwot Ridge multi-decadal warming trends in temperature and humidity (Mitton
and Ferrenberg, 2012) were included in the CLM simulations the stomatal response may have
been stronger thereby holding $c_i/c_a$ constant.

10        The simulated stomatal closure in response to $CO_2$ fertilization led to an increase in

iWUE and WUE of approximately 10-15% (Figure 6F). This is consistent with model and
observation-based studies (Ainsworth and Long, 2005; Franks et al., 2013; Peñuelas et al.,
2011) which indicate a 15-20% increase in iWUE for forests. This suggested that the
vegetation at Niwot Ridge has some ability to maintain net ecosystem productivity when
confronted with low soil moisture, low humidity conditions. Ultimately, whether Niwot Ridge
maintains the current magnitude of carbon sink (Figure 2) will depend upon the severity of
drought conditions, as improvements in WUE, in general, are only likely to negate weak to
moderate levels of drought (Frank et al., 2013).

19        The simultaneous increase in both simulated photosynthetic discrimination and iWUE

conflicts with observations where increases in iWUE are typically linked with weakening
discrimination (e.g. Saurer et al., 2004). However, under certain conditions iWUE and
discrimination can vary independently because of variation in leaf evaporative demand (VPD)
and atmospheric $CO_2$ (Seibt et al., 2008). In general, an increase in atmospheric $CO_2$ alone
tends to increase iWUE because of reduced stomatal conductance, however, the impact upon
discrimination is close to neutral because the increased supply of $CO_2$ external to the leaf is
offset by reduced stomatal conductance (Saurer et al., 2004) The VPD likely plays an important
role in determining the final trends for iWUE and discrimination, where an increasing VPD
should further reduce stomatal conductance thereby promoting the well-established relationship
(increasing iWUE, decreasing discrimination). In contrast, a weak or decreasing trend in VPD
should promote the opposite relationship (increasing iWUE, increasing discrimination). For
example, simultaneous increase in iWUE and discrimination were identified at Harvard Forest
(Belmecheri et al., 2014). Here, we do not consider multi-decadal trends in climate, therefore



increasing atmospheric $CO_2$ must be the primary driver for the simulated simultaneous increase
in discrimination and iWUE at Niwot Ridge (Figure 6).   These trends in WUE and
discrimination simulated at Niwot Ridge have also been found in a fully-coupled, isotope
enabled, global CESM simulation (Figure S2).   Specifically, a random sample of land model
grid cells representing conifer species in British Columbia (lat: 52.3° N, lon: -122.5° W) and
Quebec (lat: 49.5° N, lon: -70.0° W) all showed an increase in photosynthetic discrimination
and a 10% increase in WUE from 1850-2005.   These randomly chosen grid cells are likely
better analogs to the site-level simulations described here because they represent boreal conifer
forests, whereas the grid cells that are in the Niwot Ridge area were heterogeneous in land cover
(e.g. tundra, grassland, forest) and a poor representation of conifer forest.

11       The trends in the global simulation suggest that the site level trends are not isolated to

the specific conditions of Niwot Ridge, but are a function of the model formulation.  There is a
relationship between iWUE and $c_i/c_a$ (discrimination) as derived from equation (9) within the
CLM model,
$$\frac{c_i^*}{c_a} \cong 1 - \frac{1.6}{c_a}\ iWUE. \tag{17}$$
The full derivation is provided in the supplement.  Note that increasing iWUE is consistent with
decreasing $c_i/c_a$ ($\sim\alpha_{psn}$) and therefore consistent with established understanding between trends
in iWUE and discrimination.   However, this trend imposed by iWUE can be neutralized by
increasing  $c_a$.  During the course of the Niwot Ridge simulation iWUE increased between 10-
20% (Figure 6), however, $c_a$ increased by 40% during that same time (1850-2013).

### 3.2.2  Magnitude of photosynthetic discrimination

23       The simulated photosynthetic discrimination (Fig. 7) was significantly larger than an

estimate derived from observations and an isotopic mixing model (Bowling et al., 2014).  For
brevity we refer to the estimates based on the Bowling et al. (2014) method as 'observed'
discrimination but highlight that they are derived from observations and not directly measured.
On average, the simulated monthly growing season mean canopy discrimination was greater
than observed values by 6.3, 6.1, and 5.1‰ for the *limited nitrogen*, *unlimited nitrogen*, and *no*
*downregulation discrimination* formulations respectively. The model-observation mismatch in
discrimination, despite model-observation agreement to biomass, carbon and latent heat flux



tower observations (Figure 2) highlights the independent, and useful constraint isotopic
observations provide for evaluating model performance. Specifically, the overestimation of
discrimination may suggest the stomatal slope in the Ball-Berry model (m=9 in Eq. 4) used for
these simulations was too high. This is supported by Mao et al., (2016), who found a reduced
stomatal slope (m=5.6) was necessary for CLM 4.0 to match observed $\delta^{13}C$ in an isotope
labeling study of loblolly pine forest in Tennessee. The stomatal slope was also found to be
important to match discrimination behavior in the ISOLSM model (Aranibar et al., 2006), a
predecessor to CLM.
The mixing model approach estimate of $\Delta_{canopy}$ (17 ‰), combined with $\delta_{atm}$ (-8.25 ‰)
implies a $\delta^{13}C$ of biomass between -26 to -25 ‰. This range of values is only slightly more
enriched than the observed ranges of $\delta^{13}C$ of needle and root biomass (-27 to -26 ‰). The fact
that the different approaches to measure discrimination differ by only 1 ‰, whereas CLM
simulates a $\Delta_{canopy}$ that is 5-6 ‰ greater than the mixing model discrimination, strongly suggests
that the model has overestimated discrimination from 2006-2012. Therefore what appeared to
be a successful match between the simulated and observed $\delta^{13}C$ biomass, may in fact have been
fortuitously reached through compensating during the simulation. A multi-decadal time series
of discrimination estimates inferred from $\delta^{13}C$ of tree rings (Saurer et al., 2014; Frank et al.,
2015) would be useful to investigate this mismatch as a function of time, but these data are not
presently available.
It is likely that the overestimation of modeled discrimination originates from a lack of
response of stomatal conductance to environmental conditions. This could be a result of one or
several of the following within the model: 1) parameter calibration issue -the stomatal slope
value is too high, 2) boundary condition issue -the multi-decadal trends in climate (e.g. VPD)
have not been included in the simulation or 3) model structural issue -the Ball-Berry
representation of stomatal conductance is not sensitive enough to changes in environmental
conditions (e.g. VPD, soil moisture). It has been shown that VPD may be an improved predictor
of $g_s$ (Katul et al., 2000; Leuning, 1995) and discrimination (Ballantyne et al., 2010, 2011) as
compared to relative humidity, currently used in CLM 4.5. It would be worthwhile to clearly
identify in future work which of the three scenarios is responsible for overestimation of the
discrimination.





### 3.2.3 Seasonal pattern of photosynthetic discrimination

The model formulations that did not explicitly consider the influence of nitrogen limitation upon discrimination (*unlimited nitrogen*, *no downregulation discrimination*) were most successful at reproducing the seasonality of discrimination (Figure 7; Figure S3). In general, the observed discrimination was stronger during the spring and fall and weaker during summer. This observed $\Delta_{canopy}$ seasonal range (excluding November) varied from 16.5 to 18 ‰ using Reichstein partitioning (Figure 7), and was more pronounced using Lasslop partitioning (16.5 to 23 ‰) (Figure S3). The nitrogen limited simulated $\Delta_{canopy}$ had no seasonal trend whereas the *unlimited nitrogen* and *no downregulation discrimination* simulations both ranged from 21 to 23 ‰.

The main driver of the seasonality of discrimination was the net assimilation ($A_n$) for the *unlimited nitrogen* formulation (Figure 8). This was evident given the inversely proportional relationship between the simulated fractionation factor ($\alpha_{psn}$) and $A_n$, consistent with equation (9). Stomatal conductance ($g_s$) also influenced the seasonal pattern. The most direct evidence for this was during the period between days 175-200 (Figure 8), where $A_n$ descended from its highest value (favoring higher $\alpha_{psn}$), and $g_s$ abruptly ascended to its highest value (favoring higher $\alpha_{psn}$). The $\alpha_{psn}$ responded to this increase in $g_s$ with an abrupt increase by approximately 0.003 (3 ‰). Similarly, the *limited nitrogen* simulation seasonal discrimination pattern was shaped by both $A_n$ and $g_s$, although the magnitude for both was approximately 30% higher during the summer months as compared to the unlimited nitrogen simulation. This was because the calibrated $V_{cmax}$ value for the *limited nitrogen* simulation was much higher than for the *unlimited nitrogen* simulation (section 3.3). The difference in $\alpha_{psn}$ between the two model formulations coincided with the sharp increase in $f_{dreg}$ between days 125 and 275, providing strong evidence that the downregulation mechanism within the *limited nitrogen* formulation led to increased discrimination during the summer. Therefore, it follows that the nitrogen downregulation mechanism was the root cause of the small range in simulated seasonal cycle discrimination for the *limited nitrogen* formulation, which was inconsistent with the observations.

### 3.2.4 Environmental factors influencing seasonality of discrimination

The simulated $\Delta_{canopy}$ was driven primarily by net assimilation ($A_n$), followed by vapor pressure deficit (VPD) (Fig. 9). The correlation between VPD and $\Delta_{canopy}$ was strongest for





the *unlimited nitrogen* simulation, where the range in monthly average $\Delta_{canopy}$ spanned values
from 22 to 18 ‰ (Figure 9, middle row). This resembled the observed range in response based
upon a fitted relationship from Bowling et al., (2014) that spanned from roughly 16 to 19 ‰
(left panels of Fig. 9), although with a consistent discrimination bias. The correlation between
VPD and $\Delta_{canopy}$, however, does not demonstrate causality. If that were the case, given that $g_s$
is a function of VPD ($h_s$ term in Eq. 4) and discrimination is a function of $g_s$ (Eq. 8), a similar
relationship should have existed between $g_s$ and $\Delta_{canopy}$.    This, in fact, was not the case.
Overall, the influence of $g_s$ (responding to VPD) (R-value = -0.50) was secondary to $A_n$ (R-
value = -0.77) in driving changes in discrimination (Figure 9). The model suggested that the
range in seasonal discrimination (intra-annual variation) was driven by the magnitude of $A_n$
based on the inverse relationship between $A_n$ and $\Delta_{canopy}$, (equation 9) illustrated by the
separation between months of low photosynthesis (October, May) vs. high photosynthesis
(June, July, August). During times of relatively low photosynthesis $A_n$ also drove the inter-
annual variation in $\Delta_{canopy}$. On the other hand, $g_s$ (VPD) was most influential in driving the
inter-annual variation of discrimination during the summer months only, judging by the directly
proportional relationship during the months of June, July and August. Strictly speaking, $g_s$ is a
function of $h_s$ (leaf specific humidity) and not atmospheric VPD in CLM. However, the two
are closely related and the relationship between either variable (atmospheric VPD or simulated
leaf humidity) to $\Delta_{canopy}$ was similar (Figure S4).
The *limited nitrogen* formulation did not produce as wide a range in discrimination as
compared to the observations (Figure 9, top row). Part of this result was attributed to the lack
of response between $A_n$ and $\Delta_{canopy}$. In this case, the discrimination did not decrease with
increasing $A_n$ because the signal was muted by the countering effect of $f_{dreg}$. The *limited*
*nitrogen* formulation was, however, able to reproduce the same discrimination response to $g_s$.
as compared to the other model formulations. The tendency for the limited nitrogen model to
simulate discrimination response to $g_s$ and not to $A_n$ may negatively impact its ability to simulate
multi-decadal trends in discrimination. This may not be a major detriment to sites such as
Niwot Ridge which have maintained a consistent level of carbon uptake during the last decade,
and is likely more susceptible to environmental impact upon stomatal conductance. However,
sites that have shown a significant increase in assimilation rate (e.g. Harvard Forest; (Keenan
et al., 2013)) are less likely to be well represented by this model formulation.



Given the dependence of forest productivity at Niwot Ridge on snowmelt (Hu et al.,
2010), it was surprising that the model simulated minimal soil moisture stress (Fig. 8e) and
therefore minimal discrimination response to soil moisture. However, this finding was
consistent with Bowling et al., (2014), who did not find an isotopic response to soil moisture.
In addition, lack of response to change in soil moisture may not be indicative of poor
performance of the isotopic sub-model performance, but rather an effect of the hydrology sub-
model (Duarte et al. (in prep)). However, a comparison of observed soil moisture at various
depths at Niwot Ridge generally agrees with the CLM simulated soil moisture (not shown),
suggesting the lack of model response to soil moisture was not from biases in the hydrology
model.
**3.3  Discrimination formulations: implications for model development**
The *limited* and *unlimited model* formulations tested in this study represented two
approaches to account for nitrogen limitation within ecosystem models. The *limited nitrogen*
formulation reduced photosynthesis, *after the main photosynthesis calculation*, so that the
carbon allocated to growth was accommodated by available nitrogen. This *allocation*
*downscaling* approach is common to a subset of models, for example, CLM (Thornton et al.,
2007), DAYCENT (Parton et al., 2010) and ED2.1 (Medvigy et al., 2009). Another class of
models limits photosynthesis based upon foliar nitrogen content and adjusts the photosynthetic
capacity through nitrogen availability in the leaf though $V_{cmax}$ (e.g. CABLE, GDAY, LPJ-
GUESS, OCN, SDVGM, TECO, see Zaehle et al., 2014). These *foliar nitrogen* models are
similar to the *unlimited nitrogen* formulation of CLM because the scaling of photosynthesis
was taken into account in the $V_{cmax}$ scaling methodology (see discussion in section 2.1.2 and
2.4), *prior to the photosynthesis calculation*. In general, there were no categorical differences
in behavior between these two classes of models during $CO_2$ manipulation experiments held at
Duke forest and ORNL (Zaehle et al., 2014). CLM 4.0 was one of the few models in that study
to consistently underestimate the NPP response to an increase of atmospheric $CO_2$ due to
nitrogen limitation, however this finding was attributed to a lower initial supply of nitrogen.
The *unlimited nitrogen* formulation described in our study is a simplified foliar nitrogen
model, in that, all of the information about nitrogen limitation is incorporated within the $V_{cmax}$
downscaling approach. A more versatile approach would link a dynamic nitrogen cycle directly
with the calculation of $V_{cmax}$. This capability is currently being developed within CLM
(Ghimire et al., in review) and future work should test its functionality.





1        The performance of the *unlimited nitrogen* formulation was nearly identical to the *no*

*downregulation discrimination* formulation in terms of isotopic behavior despite the
mechanistic differences.  The *no downregulation discrimination* formulation included nitrogen
limitation within the bulk carbon behavior but ignored the impact of $f_{dreg}$ upon discrimination
behavior.  The relative high simulation skill with this formulation implied that the 'potential'
GPP linked to $A_n$, was a more effective predictor of discrimination behavior than the
'downscaled' GPP, which is linked to $A_{n*}(1\text{-}f_{dreg})$ (equation 9).  There are several potential
explanations for an unrealistically large value of $f_{dreg}$.  First this could indicate that the $V_{cmax}$
parameter was too large, thereby requiring a large $f_{dreg}$ to compensate.   As noted in Section
(3.1) the default temperate evergreen $V_{cmax25}$ was ~62 µmol m$^{-2}$ s$^{-1}$, much larger than what was
found based on literature reviews (Monson et al., 2005; Tomaszewski and Sievering, 2007) .
We found to match the observed GPP we had to impose $f_{dreg}$ that had the same effect as reducing
$V_{cmax}$ (Figure S1) to values of 51 and 34 µmol m$^{-2}$ s$^{-1}$ for the *limited nitrogen* and *unlimited*
*nitrogen* formulations respectively.  Alternatively, it could be that there are physiological
processes that are acting to reduce nitrogen limitation (e.g. nitrogen storage pools or transient
carbon storage as non-structural carbohydrates), or that the current measurement techniques are
underestimating GPP due to biases within the flux partitioning methods.
**3.4   Disequilibrium, possible explanations of mismatch**

19        Carbon cycle models (e.g. Fung et al., 1997) indicate that the steady decrease of $\delta_{atm}$ (Suess

effect, Fig. 1) should lead to a positive disequilibrium between land surface processes ($\delta^{13}$C
difference between GPP and ER, Eq. 12). This is because the $\delta_{GPP}$ reflects the most recent ($\delta^{13}$C
depleted) state of the atmosphere, whereas the $\delta_{ER}$ reflects carbon (e.g. soil carbon) assimilated
from an older ($\delta^{13}$C enriched) atmosphere.  This positive disequilibrium pattern promoted by
the Suess effect was consistent with all CLM formulations for this study with an annual average
disequilibrium of 0.8 ‰.   In contrast, a negative disequilibrium (-0.6 ‰) was identified at
Niwot Ridge based upon observations (Bowling et al. 2014) as well as in other forests (Flanagan
et al., 2012; Wehr and Saleska, 2015; Wingate et al., 2010).  Bowling et al. (2014) hypothesized
several reasons for this:  1) a strong seasonal stomatal response to atmospheric humidity, 2)
decreased photosynthetic discrimination associated with $CO_2$ fertilization, 3) decreased
photosynthetic discrimination associated with multi-decadal warming and increased VPD, and
4) post-photosynthetic discrimination.  We evaluated the first three hypotheses within the
context of the CLM simulations.



The model results suggest a seasonal variation of discrimination that is a function of both
VPD and $A_n$. The simulated seasonal range in discrimination (Figure 7; Figure S3) varied by
approximately 2 ‰, and this range in seasonal discrimination could contribute to a negative
disequilibrium provided specific timing of assimilation, assimilate storage and respiration not
currently considered in the model. For example, if a significant portion of photosynthetic
assimilation was stored during the spring with relatively high discrimination, and then respired
during the summer, the net effect would deplete the $\delta_{ER}$ and thereby promote negative
disequilibrium during the summer months when discrimination is lower. Theoretically, this
could be achieved by explicitly including carbohydrate storage pools within CLM. Isotopic
tracer studies have shown assimilated carbon can exist for weeks to months within the
vegetation and soil before it is finally respired (Epron et al., 2012; Hogberg et al., 2008).
Although carbon storage pools are included in CLM, their allocation is almost always
instantaneous for evergreen systems and could not provide the isotopic effect described above.
The $CO_2$ fertilization effect tends to favor photosynthesis in plants and has been shown to
simultaneously increase WUE and decrease stomatal conductance as inferred from $\delta^{13}C$ in tree
rings (Frank et al., 2015; Flanagan et al., 2012; Wingate et al., 2010). In general a decrease in
stomatal conductance and increase in WUE is associated with a decrease in C3 discrimination
(Farquhar et al., 1982), which opposes the disequilibrium trend imposed by the Suess effect.
The model simulation agrees with both these trends in WUE and stomatal conductance, yet
simulates an *increase* in discrimination (Figure 5; Figure 6), which reinforces the Suess effect
pattern upon disequilibrium. Although this appears to be a mismatch between forest processes
and model performance the model is operating within the limits of the discrimination
parameterization (Eq. 17) in which the magnitude of photosynthetic discrimination is inversely
proportional to the iWUE, but is also proportional to atmospheric $CO_2$ (see section 3.2.1).
A multi-decadal decrease in photosynthetic discrimination may also result from change in
climate. Meteorological measurements at Niwot Ridge during the last several decades
generally support conditions of higher VPD based upon a warming trend from an average
annual temperature of 1.1 ºC in the 1980's to 2.7 ºC in the 2000's (Mitton and Ferrenberg,
2012) and no overall trend in precipitation. It is possible that a multi-decadal trend in increasing
VPD contributed to multi-decadal weakening in photosynthetic discrimination given the
observed (Bowling et al., 2014) and modelled (Figure 9) correlation between $\Delta_{canopy}$ and VPD.
The model meteorology only included the years 1998-2013 and did not include the rapid





warming after the 1980's. It is unclear whether, if the full period of warming were to be
included in the simulation, the simulated discrimination response to VPD would be enough to
counter the Suess effect and lead to negative disequilibrium. Still, there is evidence that the
model is overestimating contemporary discrimination (Section 3.4) and the exclusion of the full
multi-decadal shift in VPD could be a significant reason why.
Finally, post-photosynthetic discrimination processes are likely to impact the magnitude
and sign of the isotopic disequilibrium (Bowling et al., 2008; Brüggemann et al., 2011) at
multiple temporal scales. None of these isotopic processes are currently modelled within CLM
4.5, so at present the model cannot be used to examine them.
**4  Conclusions**
This study provides a rigorous test of the representation of C isotope discrimination within
the highly mechanistic terrestrial carbon model CLM. Special attention was paid to provide an
accurate set of boundary conditions to isolate the isotopic performance including 1) customized
atmospheric $CO_2$ and $\delta_{atm}$ time series, 2) customized model initialization procedure, and 3)
empirical $V_{cmax}$ calibration procedure. Once the model satisfactorily represented observed
carbon exchange, water exchange, and biomass growth, it was successful at simulating several
aspects of isotope behavior.
CLM was able to accurately simulate $\delta^{13}C$ in leaf and stem biomass and the seasonal cycle
in $\Delta_{canopy}$. This performance could only be achieved, however, if $V_{cmax}$ were calibrated in such
a way to mimic the functionality of a foliar nitrogen model by accounting for nitrogen limitation
*prior* to photosynthesis. With the traditional *nitrogen limited* approach, in which the nitrogen
limitation occurs *after* photosynthesis and the $c_i/c_a$ is influenced by this limitation but the
stomatal conductance is not, the model tended to overestimate the magnitude of photosynthetic
discrimination, and eliminated the observed seasonal weakening of $\Delta_{canopy}$. Although the
overestimation of photosynthetic discrimination could likely be corrected with adjustments to
the stomatal conductance parameterization, the seasonal trend was inherent to the model. Thus
our results suggest that shifting nitrogen controls either before photosynthesis through a
reduction in $V_{cmax,}$ or entirely after the photosynthetic process such that nitrogen constraints
have no effect on discrimination, are more consistent with the isotopic observations than the
current model formulation.
Although the *unlimited nitrogen* formulation was able to match observed $\delta^{13}C$ of biomass
and seasonal patterns in discrimination, it still overestimated the contemporary magnitude of



discrimination (2006-2012). Future work should identify whether this overestimation was a
result of parameterization (stomatal slope), exclusion of multi-decadal shifts in VPD, or
limitations in the representation of stomatal conductance (Ball-Berry model).
The model attributed most of the range in seasonal discrimination to variation in net
assimilation rate ($A_n$) followed by variation in VPD, with little to no impact from soil moisture.
The model suggested that $A_n$ drove the seasonal range in discrimination (across-month
variation) whereas VPD drove the inter-annual variation during the summer months. This
finding suggests that to simulate multi-decadal trends in photosynthetic discrimination,
response to assimilation rate and VPD must be well represented within the model.
The model simulated a positive disequilibrium that was driven by both the Suess effect,
and increased photosynthetic discrimination from $CO_2$ fertilization. It is possible that the
negative disequilibrium that was inferred from observations (Bowling et al., 2014) was driven
from the impacts of climate change and/or post-photosynthetic discrimination – not considered
in this version of the model. Future work should quantify the impact of this multi-decadal
warming and post-photosynthetic discrimination processes upon disequilibrium.
The model simulated a consistent increase in water-use efficiency as a response to $CO_2$
fertilization and decrease in stomatal conductance. The model simulated an increase in WUE
despite an increase in discrimination, however C3 plants typically express the opposite trends
(increase in WUE, decrease in discrimination). Although CLM includes parameterization that
promotes an increase in WUE with a decrease in discrimination, this trend was likely
neutralized by other environmental variables (e.g. increase in $c_a$).
Initial indications are that $\delta^{13}C$ isotope data can bring additional constraint to model
parameterization beyond what traditional flux tower measurements of carbon, water exchange,
and biomass measurements. The isotope measurements suggested a stomatal conductance
value generally lower than what was consistent with the flux tower measurements.
Unexpectedly, the isotopes also provided guidance upon model formulation related to nitrogen
limitation. The success of our empirical approach to account for nutrient limitation within the
$V_{cmax}$ parameterization, suggests that additional testing of foliar nitrogen models are
worthwhile.





## 1   **Acknowledgements**

This research was supported by the U.S. Department of Energy, Office of Science, Office of
Biological and Environmental Research, Terrestrial Ecosystem Science Program under Award
Number DE-SC0010625. Thank you to Sean Burns and Peter Blanken for sharing flux tower
and meteorological data from Niwot Ridge. Thank you to those at NOAA who provided the
atmospheric flask data from Niwot Ridge including Bruce Vaughn, Ed Dlugokencky, the
INSTAAR Stable Isotope Lab and NOAA GMD. A special thanks to Keith Lindsay at NCAR
for providing global CESM output to help improve the discussion of model behavior. The
support and resources from the Center for High Performance Computing at the University of
Utah are gratefully acknowledged.





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



1    Table 1.   List of symbols used.

| Symbol | Description | Unit or Unit Symbol |
|---|---|---|
| $\alpha_{psn}$ | Fractionation factor ($R_a$/ $R_{GPP}$) | dimensionless |
| $\beta_t$ | Soil water stress parameter (BTRAN) | dimensionless |
| $\Delta_{canopy}$ | photosynthetic carbon isotope discrimination | ‰ |
| $\delta^{13}C$ | $^{13}C/^{12}C$ isotope composition (relative to VPDB) | ‰ |
| $\delta_{atm}$ | $\delta^{13}C$ of atmospheric $CO_2$ | ‰ |
| $\delta_{ER}$ | $\delta^{13}C$ of ecosystem respiration | ‰ |
| $\delta_{GPP}$ | $\delta^{13}C$ of net photosynthetic assimilation | ‰ |
| $\Gamma_*$ | $CO_2$ compensation point | Pa |
| $A_c$ | Enzyme-limiting rate of photosynthetic assimilation | $\mu mol\ m^{-2}\ s^{-1}$ |
| $A_j$ | Light-limiting rate of photosynthetic assimilation | $\mu mol\ m^{-2}\ s^{-1}$ |
| $A_p$ | Product-limiting rate of photosynthetic assimilation | $\mu mol\ m^{-2}\ s^{-1}$ |
| $A_n$ | net photosynthetic assimilation | $\mu mol\ m^{-2}\ s^{-1}$ |
| $Resp_d$ | Leaf-level dark respiration | $\mu mol\ m^{-2}\ s^{-1}$ |
| $a_{R25}$ | Specific activity of Rubisco at 25ºC | $\mu mol\ g^{-1}\ Rubisco\ s^{-1}$ |
| $b$ | Minimum stomatal conductance | $\mu mol\ m^{-2}\ s^{-1}$ |
| $CF_{alloc}$ | Actual carbon allocated to biomass (N-limited) | $gC\ m^{-2}\ s^{-1}$ |
| $CF_{av\_alloc}$ | Maximum carbon available for allocation to biomass | $gC\ m^{-2}\ s^{-1}$ |
| $CF_{GPPpot}$ | Potential gross primary production (non N-limited) | $gC\ m^{-2}\ s^{-1}$ |
| $c_a$ | Atmospheric $CO_2$ pressure | Pa |
| $c_i$ | Leaf intracellular $CO_2$ pressure | Pa |
| $c_i*$ | Leaf intracellular $CO_2$ pressure, (N-limited) | Pa |
| $c_s$ | Leaf surface $CO_2$ pressure | Pa |
| $e_l$ | Saturation vapor pressure | Pa |
| $e_s$ | Water vapor pressure at leaf surface | Pa |
| $E_T$ | Leaf Transpiration | $\mu mol\ m^{-2}\ s^{-1}$ |
| ER | Ecosystem respiration | $\mu mol\ m^{-2}\ s^{-1}$ |
| GPP | Gross primary productivity (photosynthesis) | $\mu mol\ m^{-2}\ s^{-1}$ |
| $F_{LNR}$ | Fraction of leaf nitrogen within Rubisco | $gN\ Rubisco\ g^{-1}\ N$ |
| $F_{NR}$ | Total Rubisco mass per nitrogen mass within Rubisco | $g\ Rubisco\ g^{-1}\ N\ Rubisco$ |
| $f_{df}$ | $V_{cmax}$ scaling factor | dimensionless |
| $f_{dreg}$ | Nitrogen photosynthetic downregulation factor | dimensionless |
| $g_b$ | Leaf boundary layer conductance | $\mu mol\ m^{-2}\ s^{-1}$ |
| $g_s$ | Leaf stomatal conductance | $\mu mol\ m^{-2}\ s^{-1}$ |
| $h_s$ | Leaf surface humidity | $Pa\ Pa^{-1}$ |
| $K_c$ | Michaelis-Menten constant | Pa |
| $K_o$ | Michaelis-Menten constant | Pa |
| LE | Latent heat flux | $W\ m^{-2}$ |
| $m$ | Stomatal slope (Ball Berry conductance model) | dimensionless |
| $Na$ | Leaf nitrogen concentration | $gN\ m^{-2}\ leaf\ area$ |
| NEE | Net ecosystem exchange | $\mu mol\ m^{-2}\ s^{-1}$ |
| NPP | Net primary production | $\mu mol\ m^{-2}\ s^{-1}$ |
| $o_i$ | $O_2$ atmospheric partial pressure | Pa |
| PFT | Plant functional type | N/A |
| $P_{atm}$ | Atmospheric pressure | Pa |
| $R_a$ | Isotopic ratio of canopy air | $^{13}C/^{12}C$ |

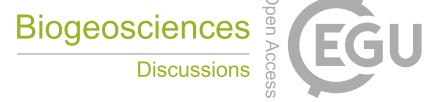



| $R_{GPP}$ | Isotopic ratio of net photosynthetic assimilation | $^{13}C/^{12}C$ |
|---|---|---|
| $R_{VPDB}$ | Isotopic ratio of Vienna Pee Dee Belemnite standard | $^{13}C/^{12}C$ |
| $r$ | Fraction of roots (for $\beta_t$) | dimensionless |
| $V_{cmax25}$ | Maximum carboxylation rate at 25°C | $\mu mol\ m^{-2}\ s^{-1}$ |
| $V_{cmax}$ | Maximum carboxylation rate at leaf temperature | $\mu mol\ m^{-2}\ s^{-1}$ |
| VPD | Vapor pressure deficit | Pa |
| $w$ | Plant wilting factor (for $\beta_t$) | dimensionless |
| $WUE$ | Water use efficiency, ground area basis | $gC\ gH_2O^{-1}$ |
| $iWUE$ | Intrinsic water use efficiency, leaf area basis | $gC\ gH_2O^{-1}$ |



1    Table 2. CLM 4.5 model formulation description based upon timing of nitrogen limitation.
2    Pre-photosynthetic and post-photosynthetic nitrogen limitation are achieved through $V_{cmax25}$
3    calibration (equation 15) and $f_{dreg}$ (equation 7) respectively.

| Formulation | Pre-Photosynthetic Nitrogen Limitation | Post-Photosynthetic Nitrogen Limitation | Impact on $c_i/c_a$ & discrimination | Impact on stomatal conductance |
|---|---|---|---|---|
| *Limited nitrogen (default)* | Yes (weak) | Yes, $f_{dreg} > 0$ | Yes | No |
| *Unlimited nitrogen* | Yes (strong) | No, $f_{dreg} = 0$ | Yes | Yes |
| *No downregulation discrimination* | Yes (weak) | Yes, $f_{dreg} > 0$ | No | No |



Table 3. CLM 4.5 key parameter values for all model formulations

| Parameter | Description | Value | Units |
|---|---|---|---|
| *froot_leaf* | new fine root C per new leaf C | 0.5 | gC gC$^{-1}$ |
| *froot_cn* | fine root (C:N) | 55 | gC gN$^{-1}$ |
| *leaf_long* | leaf longevity | 5 | years |
| *leaf_cn* | leaf (C:N) | 50 | gC gN$^{-1}$ |
| *lflitcn* | leaf litter (C:N) | 100 | gC gN$^{-1}$ |
| *slatop* | specific leaf area (top canopy) | 0.007 | m$^2$ gC$^{-1}$ |
| *stem_leaf* | new stem C per new leaf C | 2 | gC gC$^{-1}$ |
| *mp* | stomatal slope | 9 | |
| *croot_stem* | coarse root: stem allocation | 0.3 | gC gC$^{-1}$ |
| *deadwood_cn* | dead wood (C:N) | 500 | gC gN$^{-1}$ |
| *livewood_cn* | live wood (C:N) | 50 | gC gN$^{-1}$ |
| *flnr* | fraction of leaf nitrogen within Rubisco enzyme | 0.0509 | gN gN$^{-1}$ |
| *decomp_depth_e_folding* | controls soil decomposition rate with depth | 20 | m |



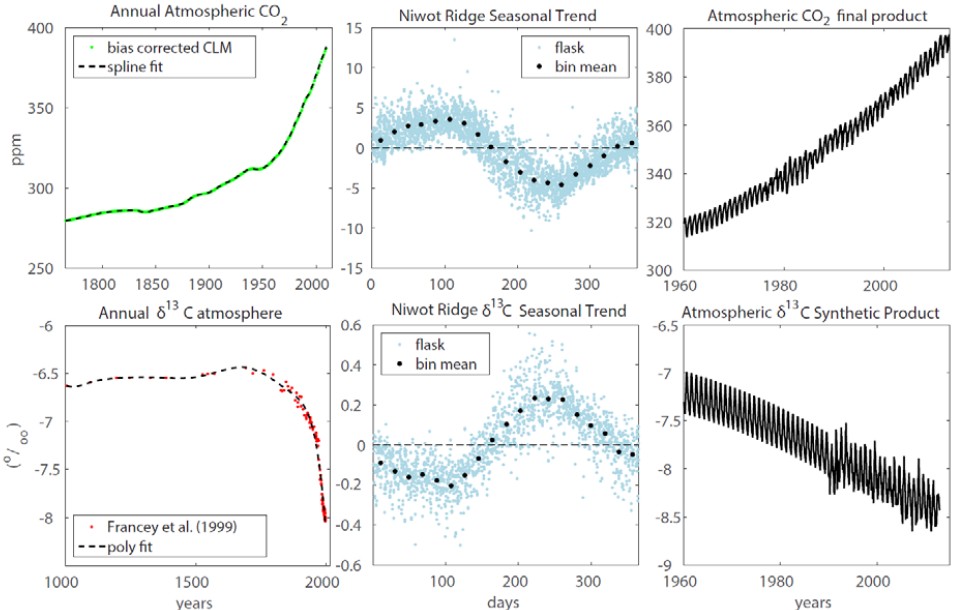

Figure 1. Niwot Ridge synthetic data product for atmospheric $CO_2$ concentration ($c_a$) (top row)
and $\delta^{13}C$ of $CO_2$ ($\delta_{atm}$) (bottom row). The final time series (right column) was used as a
boundary condition for CLM, and created by combining the annual trends reported by Francey
et al. (1999) adjusted for Niwot Ridge (left column) with the mean seasonal cycles measured at
Niwot Ridge (middle column).



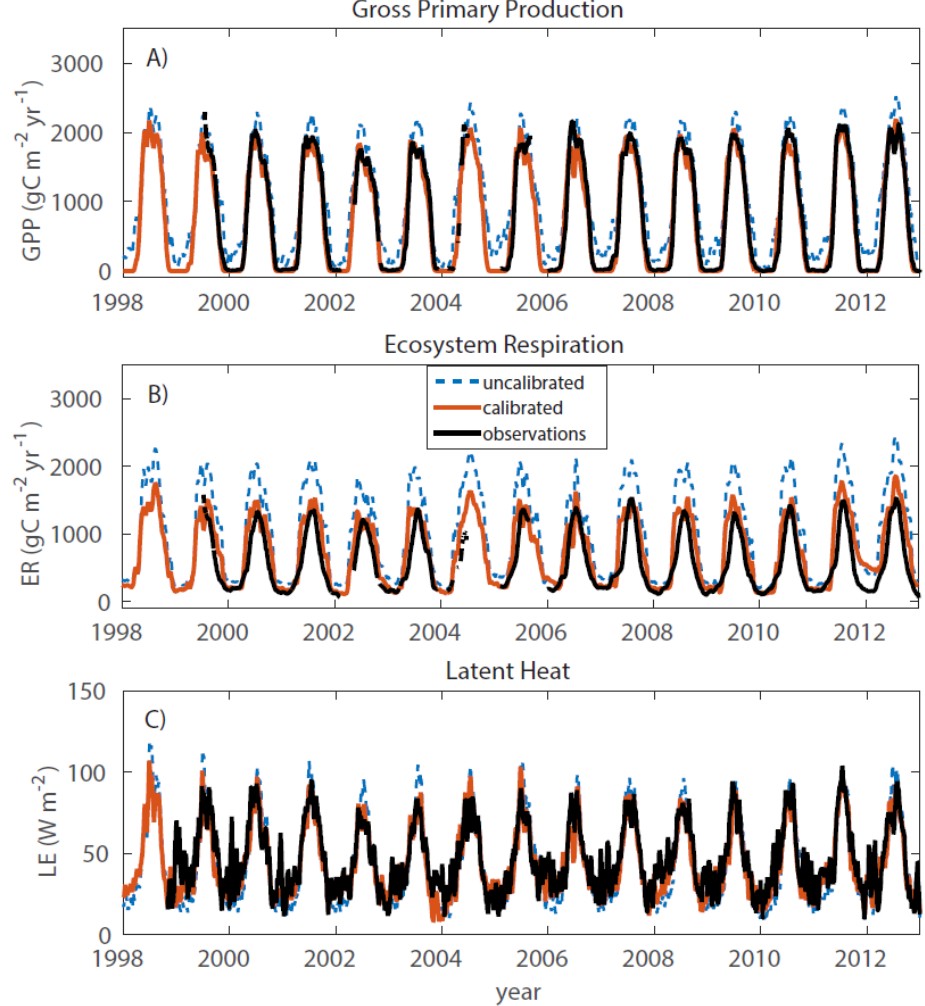

Figure 2. Simulated and observed land-atmosphere fluxes of A) gross primary production

(GPP) B) ecosystem respiration (ER) and C) latent heat (LE) for the *limited nitrogen* simulation.

The 'observations' are taken from the Ameriflux L2 processed eddy covariance flux tower data,

partitioned into GPP and ER using the method of Reichstein et al. (2005). The *uncalibrated*

simulation represents the CLM simulation without $V_{cmax}$ scaling and the *calibrated* simulation

represents the CLM run using the $V_{cmax}$ scaling approach.



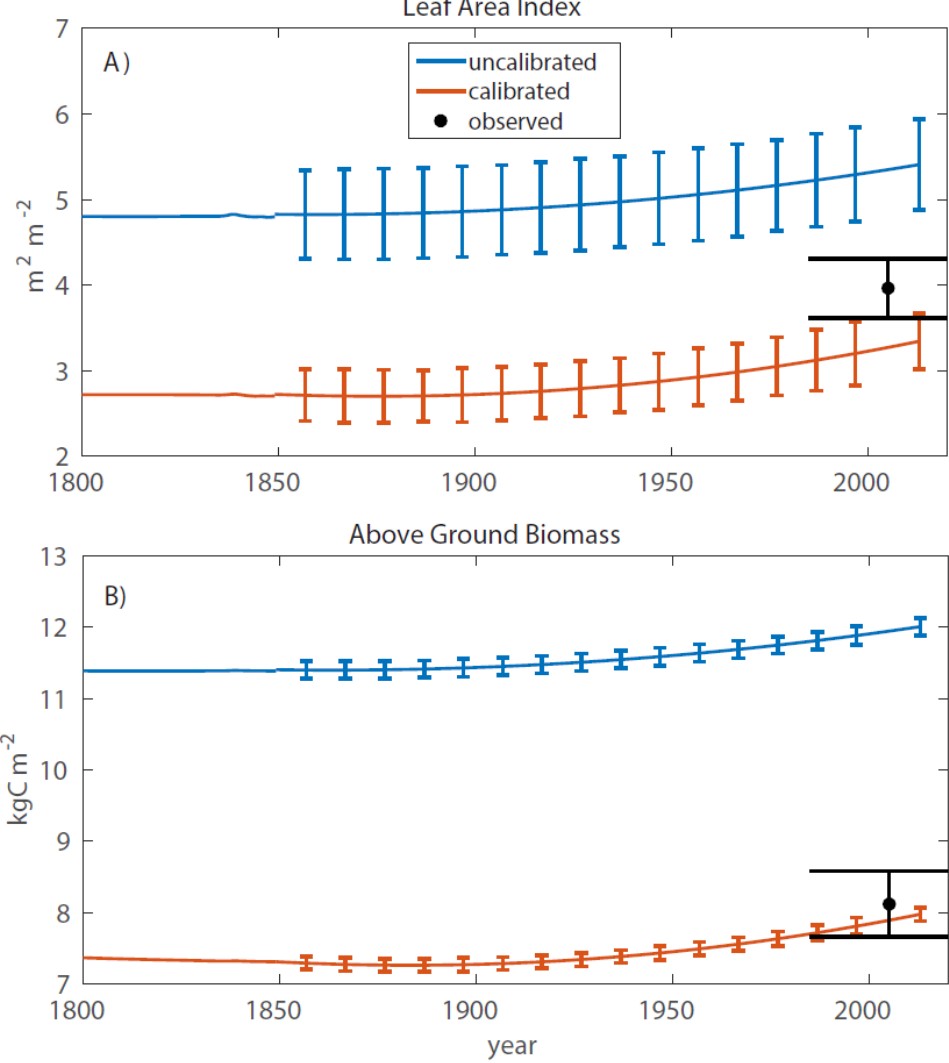

Figure 3. Simulation of A) leaf area index and B) above ground biomass for both uncalibrated
and calibrated ($V_{cmax}$ downscaled, *limited nitrogen*) simulation.  Observations are from
Bradford et al. (2008) with uncertainty bars representing standard error.  Uncertainty bars on
simulated runs represent 95% confidence of biomass variation as a result of cycling the site
level meteorology observations.



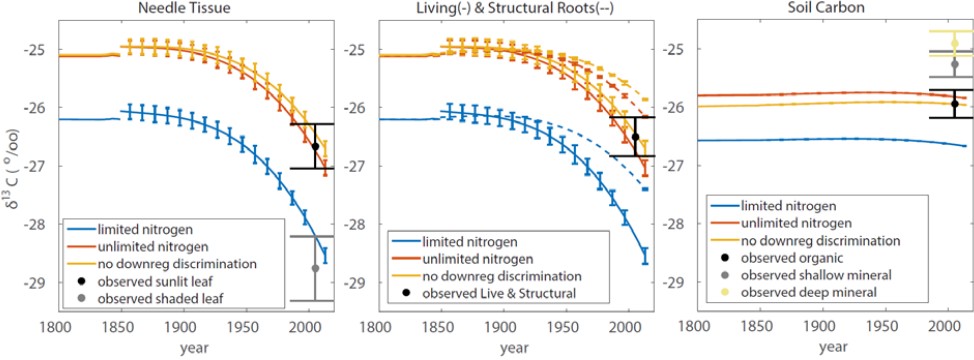

Figure 4. Simulation of $\delta^{13}C$ of bulk needle tissue, bulk roots and bulk soil carbon. A
description of model formulations are provided in Table (2). Uncertainty bars for simulations
represent 95% confidence intervals of $\delta^{13}C$ variation as a result of cycling the site level
meteorology observations. The observed values are from Schaeffer et al. (2008) with
uncertainty bars representing standard error. Solid lines and dashed lines in middle panel
represent living roots and structural roots respectively.

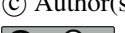


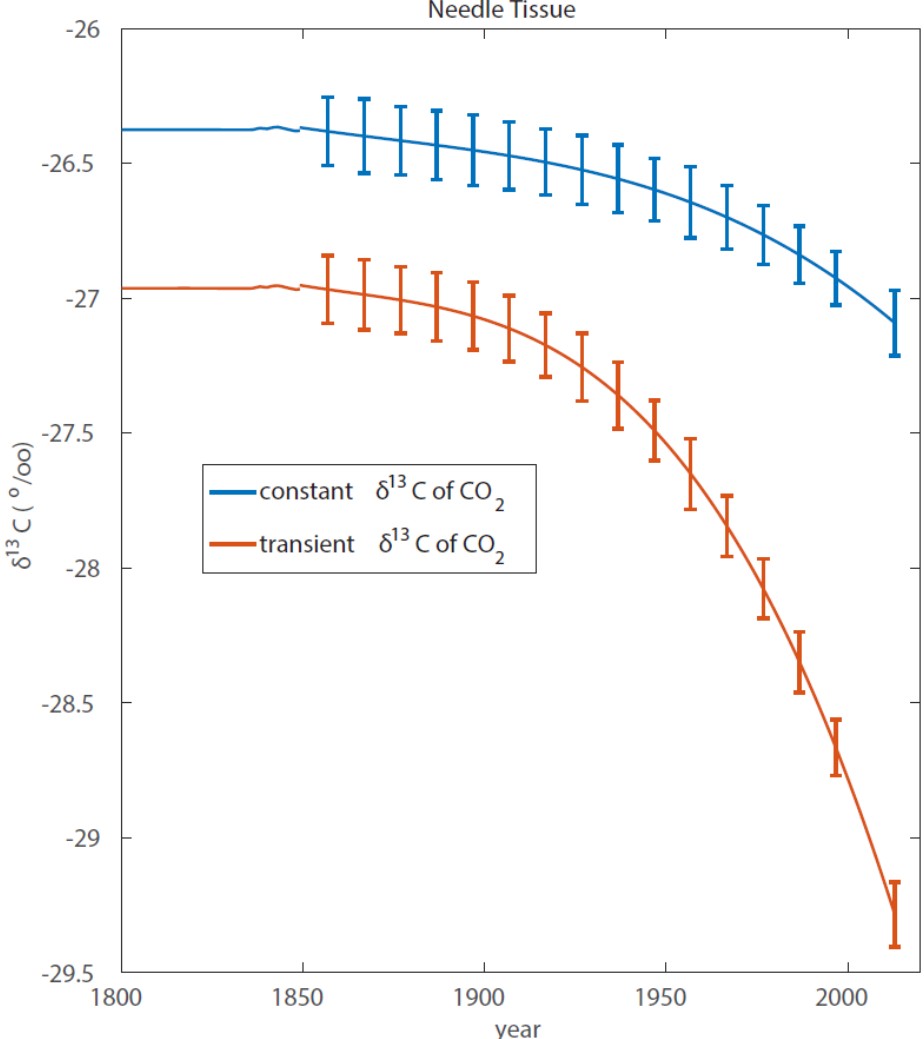

2    Figure 5.   Simulation of $\delta^{13}$C of needle tissue using the *limited nitrogen* (default) CLM run. In

3    the *constant $\delta^{13}$C of CO2 ($\delta_{atm}$)* simulation the model boundary condition was -6 ‰, whereas

4    the *transient $\delta_{atm}$* simulation varied over time (Figure 1).





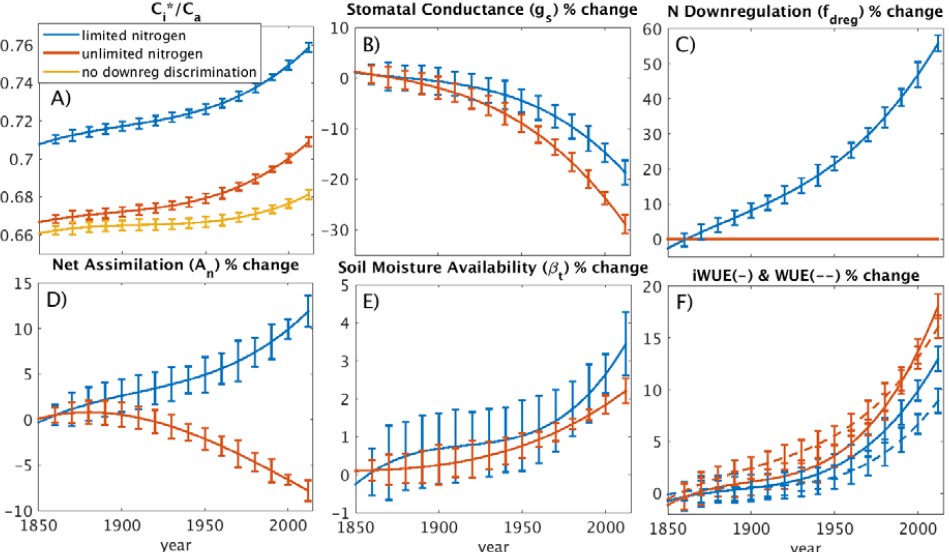

Figure 6. Diagnostic model variables that explain the discrimination trends (Figure 4) for the three model formulations as described in Table (2) for A) $c_i*/c_a$, B) $g_s$, C) $f_{dreg}$, D) $A_n$, E) $\beta_t$, and F) the water use efficiency (WUE) and intrinsic water use efficiency (iWUE). Where the *no downregulation discrimination* simulation is not shown, it was identical to the *limited nitrogen* simulation. Uncertainty bars represent 95 % confidence intervals of diagnostic variable variation as a result of cycling the site level meteorology observations. The dashed lines represent WUE and the solid lines represent iWUE in panel F.




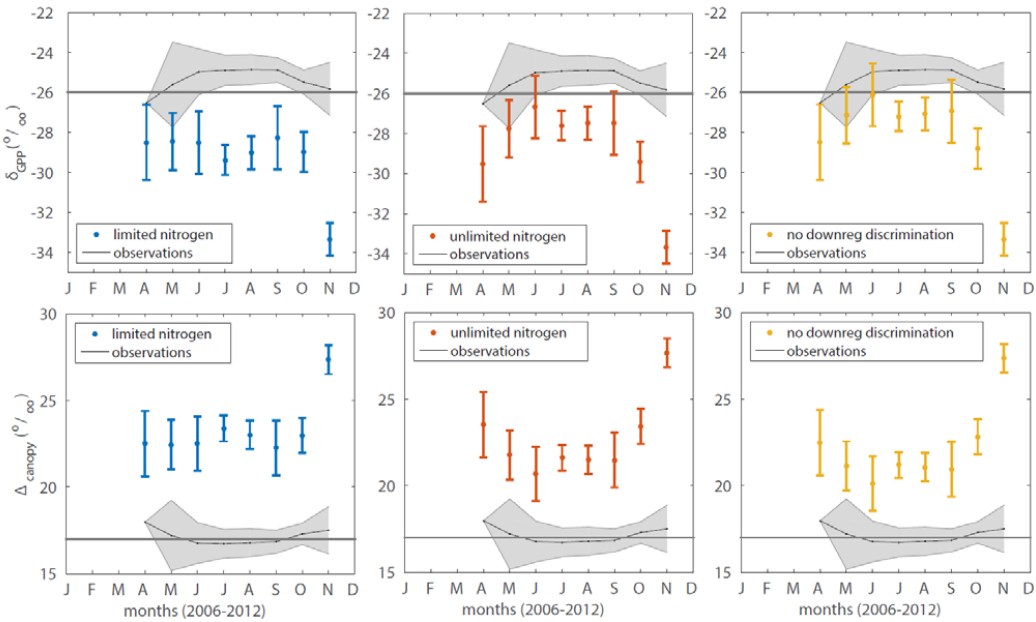

Figure 7. The seasonal pattern of photosynthetic discrimination as shown through $\delta_{GPP}$ (top
row) and $\Delta_{canopy}$ (bottom row). Uncertainty bars represent 95% confidence bounds of simulated
monthly average values from 2006-2012. Gray-shaded observation bounds represent 95%
confidence intervals of 'observed' monthly average values based upon isotopic mixing model
using Reichstein et al. (2005) partitioning of net ecosystem exchange flux described by
(Bowling et al. 2014). The horizontal lines at $\delta^{13}C$ of -26 ‰ (top row) and 17 ‰ (bottom row)
are included for reference.





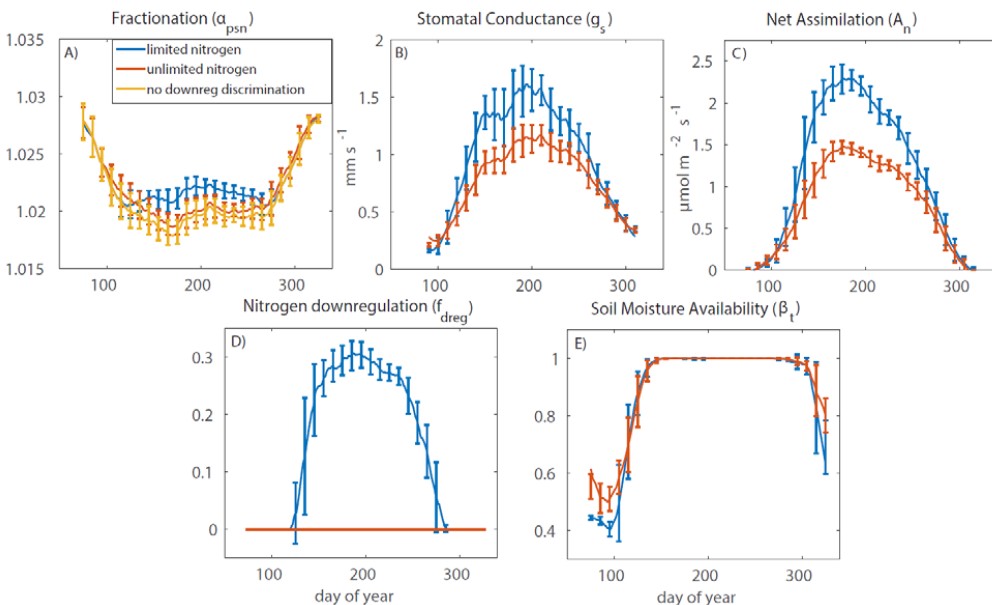

Figure 8. The seasonal pattern of discrimination (panel A) and diagnostic variables that explain
the discrimination pattern in Figure (7).   The individual tiles provide behavior from days 75-
325 for A) $\alpha_{psn}$, B) $g_s$, C) $A_n$, D) $f_{dreg}$, and E) $\beta_t$. Where the *no downregulation discrimination*
model simulation is not shown, it is identical to the *limited nitrogen* simulation.   Uncertainty
bars represent 95 % confidence intervals of inter-annual variation from 2006-2012.





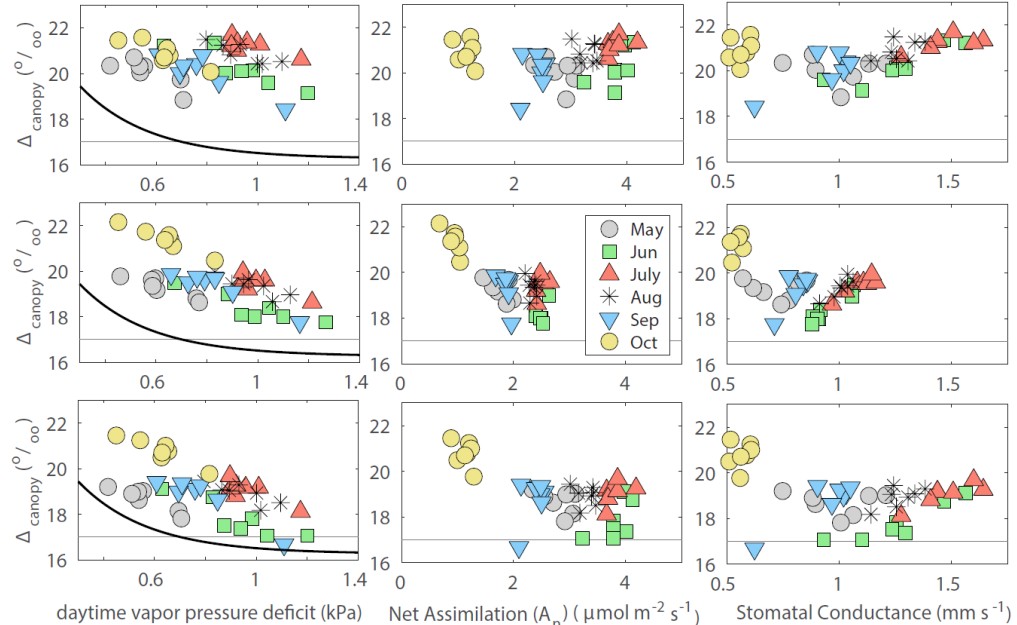

Figure 9.   Relationship between monthly average photosynthetic discrimination and monthly
average vapor pressure deficit (1st column), $A_n$ (2nd column) and $g_s$ (3rd column) from 2006-
2012.  The rows represent the *limited nitrogen* (row 1), *unlimited nitrogen* (row 2), and *no*
*downregulation discrimination* (row 3) simulations.  The black line in the 1st column is based
on exponential fitted line from observed relationship at Niwot Ridge (Bowling et al. 2014).  The
horizontal lines represent $\delta^{13}C$ of 17 ‰ and are included for reference.

