# Peer review of "Figure S1. Calibrated and uncalibrated simulations for the *limited nitrogen* formulation (top panel) and *unlimited nitrogen* formulation (bottom panel). The *limited nitrogen* calibrated run used the $V_{cmax25}$ calibration parameter: (equation (A1)) $f_{df} = (1.09 * 10^{-12})x^6 - (1.351 * 1"

_Biogeosciences, 2016_

## Short Comment (SC1) · 1 Apr 2016

Overall, very nice paper, which I found quite educational. I'm passing on here a few comments that I jotted down as I read through the paper. Most of these relate to communicating the framework by which nitrogen limitation is implemented in CLM. Admittedly, I didn't go back and consult the prior references. But I'm imagining a clearer presentation of the key concepts might be possible without doing this. The presentation in the paper gave me only a fuzzy gist. Perhaps it would be possible (?) to add a figure which diagrams the carbon flows from the atmosphere, through stomata to

substrate formation for each of the three formulations. I'm imagining that the diagram would have arrows for each of these quantities: An, GPP, CFavailable_alloc, CF_alloc, CF_GPPpot, etc. Or maybe one figure would suffice, assuming the knobs to switch between formulations is clear enough. For example, where is the carbon that is fixed but not allocated ending up? Is it respired? If so, does this respiration return back from the stomata or return through some other pathway?

Page 6, line 28: I'm missing how An is related to terms in Eq. (6). It would very much help to include an algebraic expression for this.

Page 6, lines 29-30. From the wording it sounds like maintenance respiration is partly double counted. Page 7, line 11. What does the subscript psn signify? Perhaps could be omitted? Page 7, line 15: This formula suggests that An is not equal to the flux through stomata. So what is An equal to? Is it the same as potential photosynthesis? If so, needs stating. See earlier comment also.

Page 7, lines 22-23: This sentence is a bit ambiguous. Are both given in Eq. 9, or just one. If not both, then how is nitrogen limitation incorporated? Reading below, I see this is probably related to control of Vcmax. If so, this need stating more clearly earlier, i.e. how does nitrogen limitation influence Na in Eq. (3)?

Page 8, line 22. I'm missing an expression for how delta_GPP is calculated from alpha_psn. (Okay, I know enough to work this out for myself, but I'm not sure you should assume all readers would).

Page 8, line 28. It would seem important to clarify is meant here by GPP. Which of these is it: An, CFavailable_alloc, CFGPP_pot, etc. ? Also, Perhaps I missed it, but I think the symbol E_T hasn't been defined.

---

## Short Comment (SC2) · 2 Apr 2016

Thank you for your comments. I am pleased that you found the paper informative. Land surface models such as CLM can be very detailed and complex, so it is always a challenge to know just how much of the detail to provide in the text and how much to leave to the citations. In this case, most of your concerns can be answered from the technical description of CLM 4.5 in Oleson et al. (2013), which we cite in the text.

Nevertheless, it is important to make manuscripts as transparent as possible, and agree that an overall diagram summarizing the key assimilation and allocation steps

would be helpful here (see attached figure as an example of what we intend to include in a revised version).

I am confident we can address your other line item concerns in the revised text as well.

Brett Raczka

—————————————————————

[Figure]

[Figure]

Figure 1: A simplified representation for CLM 4.5 of assimilation and allocation of carbon for conifer species. Colored boxes and arrows represent carbon pools and carbon fluxes respectively. Clear background boxes represent CLM sub-models. The N-limitation is determined by required N availability to meet demand from C:N ratio based on $CF_{Avail\_alloc}$. The blue and red text and arrows represent the limited and unlimited nitrogen formulations respectively. The no-downregulation discrimination formulation is exactly the same as the limited N formulation.

**Fig. 1.** Overview of carbon assimilation and allocation

---

## Referee Comment (RC1) · Anonymous Referee #2 · 7 Apr 2016

Raczka et al. present an interesting attempt to link carbon isotope data with the function of CLM4.5. I have listed a few thoughts below:

- I found the introduction very clear but I wonder if there is any other literature on how other models have used isotope data? I realise the authors suggest this is the first time it has been attempted in CLM and I realise this paper is primarily targeted at the CLM community, nevertheless I think my one concern would be the lack of literature in relation to other models and isotopes? Currently it is largely non-existent.

- Equation 1: I don't think you mean Respd = dark respiration. Rdark is not the same

as day respiration/respiration in the light. Suggest the use of Rday or Rd.

- Equation 4: I'm pretty sure that "Bt" should be applied to your slope term "m", rather than the minimum stomatal conductance, b? Can you please check you have this correct. Certainly if you do then this would be a striking departure from most other models.

- Line 23: "tree canopy" is this only true for trees, what happens with grasses in the model? If not, perhaps delete tree and leave just canopy.

- Equation 8 & 9: it would be helpful to the reader to explain where the numbers 4.4, 22.6 and 1000 come from, or what conversions they apply to.

- Century model (line 26/27) should have a reference.

- I'm not sure what the length of the paper was but the results/discussion text did feel very long? Similarly the conclusions runs to nearly two pages. This seems excessive to me. I'm fairly confident there are cuts that could be made to the text which would make it more digestible to the reader. I certainly found myself losing track during my reading and I think this is the key area which requires editing during revision.

- The authors note: "the overestimation of discrimination may suggest the stomatal slope in the Ball-Berry model (m=9 in Eq. 4) used for these simulations was too high." While it is may be true that the slope parameter is poorly informed by site data, the logic of this conclusion in itself may not be valid. Isotopic measurements *should* give lower slope values than those one would infer via leaf gas exchange data (i.e. the data used to inform the Ball-Berry model). This is because leaf gas exchange measures the resistance from the intercellular spaces (Ci), whereas isotopes measures the resistance from the chloroplast (Cc). I see no mention of this in the text and caution against the authors potential drawing the wrong conclusion from the model-data discrepancy.

- In discussing the "limited nitrogen formulation", the authors note: "In general, there were no categorical differences in behavior between these two classes of models dur-

[Figure]

Interactive
comment

ing CO2 manipulation experiments held at Duke forest and ORNL (Zaehle et al., 2014). CLM 4.0 was one of the few models in that study to consistently underestimate the NPP response to an increase of atmospheric CO2 due to nitrogen limitation, however this finding was attributed to a lower initial supply of nitrogen." This is not strictly true. As part of the same model-data inter-comparison of the models to the data at the two FACE sites, De Kauwe et al. (2013, Global Change Biology), found no support for the implementation whereby assimilation is limited by nitrogen availability, but not stomatal conductance. They concluded: "Stomatal conductance data from both sites were used to test modelled leaf-level responses. The simple stomatal conductance model (Eq. 1) fitted the data well (Fig. 6), supporting the assumption of coupling between assimilation and stomatal conductance. Importantly, at the ORNL site, N content of the foliage declined strongly over the course of the experiment (Norby et al., 2010), but neither the slope of the stomatal model, nor the response of A/gs to CO2, was altered by this decline (Fig. 6b). These data indicate that the coupling between stomatal conductance and assimilation is not affected by N-limitation (Fig. 6b). The data therefore tend to support coupled models over uncoupled, or partially coupled, models such as DAY-CENT and CLM4." Furthermore, I would question if there is any evidence that plants follow the "limited nitrogen formulation"?

- Phrases like "The relative high simulation skill" or "CLM was able to accurately" need some quantification. There are a number of similar cases dotted around the manuscript.

- Figure 2. I realise that a strength of this paper is the long timeseries; however, showing ∼15 years of data like that isn't particularly instructive. It is hard to distinguish the model-obs differences. Perhaps average a day/week or monthly climatology across years would more clearly show differences. This figure could also be kept, perhaps one could go to the supplementary.

- Figure 8e. I find it hard to believe that there is no reduction in the soil moisture availability factor during the whole of the summer? This seems unlikely to me? Could

this please be checked?

- Figure 9. I would suggest the symbol sizes could be reduced, they seem a little large for the figure panels.

---

## Referee Comment (RC2) · Anonymous Referee #1 · 18 Apr 2016

The manuscript by Raczka et al. provides an extensive description of how carbon isotope discrimination is represented in the CLM land surface model and how it can be used as a constraint for evaluating model performance. The model evaluation conducted in this paper provides valuable insights into the representation of physiological processes in process-based models, including potential improvements with regard to model structure and parameterization. This information will be useful to the wider land surface modeling community. Further improvements to the manuscript should include a few clarifications, in particular on how N limitation is implemented in the model. I have listed detailed comments here:

[Figure]

- The abstract could explicitly mention that three different N-limitation formulations were tested in the model. This information is worth to mention, but somewhat hidden in the abstract. E.g. it would make sense to shortly explain what the "alternative nitrogen limitation" formulation in line 29 actually means. To compensate for the additional number of words one could shorten the Vcmax calibration description or try to focus on a few key outcomes.

- The original source of the Ball Berry model (Ball et al. 1987) should be acknowledged. Further, hs represents relative, and not specific humidity at the leaf surface. The definition of hs is unnecessary.

- ci is intercellular, not intracellular $CO_2$ partial pressure, unless you consider mesophyll conductance, which seems not to be the case.

- Equation 13: what does ET represent? In Table 1 it is listed as leaf transpiration, but that clearly doesn't make sense here. But I wonder if it is ecosystem transpiration or evapotranspiration?

- Table 1: please check the unit for iWUE, it shouldn't be gC $gH2O-1$. Add $CO_2$ and $O_2$ for the Michaelis Menten constants.

- In Equation 14, I assume you mean "An" rather than "A". Please clarify.

- Please provide latin names for the dominant species at the site.

- Equation 8: please state where the 4.4 and 22.6 come from and which of those represents fractionation due to diffusion and Rubisco. I am not sure if this is clear to all readers.

- 2.1.2: The comparison of different versions of how nitrogen limitation is implemented in the model and its implications is a very interesting aspect covered by the manuscript. Unfortunately, the three different formulations tested (unlimited N, limited N, no down-regulation discrimination) are described in a rather confuse way, and I doubt that it will comprehensible for all readers. I strongly encourage the authors to include the

overview figure that they have shown in an earlier comment and I recommend a better explanation of Equation 7. How is N-limitation determined? This is mentioned in the Figure caption, but one could also include this in the manuscript as well. Further, the terms "potential" and "actual photosynthesis" are mentioned on page 7, line 17f, but they haven't been defined before, and they aren't common terms either. In the standard (= limited nitrogen) version, is photosynthesis first calculated without N-limitation, then N-limitation calculated according to Equation 7, and then the actual photosynthesis calculated by An*(1-fdreg)? But then, how is it possible that a reduction in An caused by N-limitation does not feedback on gs? This should be the case considering Eq. 4. The approach becomes clearer after reading section 3.3, but it would be helpful to explain it better at this point.

- 2.2 State here that NEE and other fluxes are observations based on the eddy covariance method. Please clarify here that the NEE partitioning was conducted using two different methods, and briefly mention their approach.

- P.10 line 27ff: that's a very detailed description which seems unnecessary to me. One could shorten this part or omit completely.

- Same is true for the last sentence in 2.3 and the first sentences of 2.4., where many technical and CLM-specific details are mentioned that one may consider to omit, as they are of lesser interest to the wider community.

- 3.1.1 what caused the overestimation of LAI, GPP etc. in the uncalibrated simulation? I presume the default value of Vcmax was too high?

- Figure 2: I think it would make more sense to show a mean annual course of the three variables rather than the complete time series. The way it is now makes it hard to see by how much GPP, ER, and LE differ from the observations on average. An interesting aspect is the underestimation of WUE. Is this more related to evaporation or transpiration? In the latter case this would be strongly related to the stomatal slope parameter "m" in the Ball-Berry model (see later comment), but could have other reasons as well.

One could shortly comment on this, up to the authors.

- P.15 line 15: the fact that stomatal conductance responds to atmospheric $CO_2$ is long known. I suggest citing an earlier study that showed this.

- P.16 line 13: Please make sure that the iWUE trend reported in the studies cited here refer to the same time period. Over which timespan did the 15-20% increase in iWUE occur according to these studies?

- 3.2.1 you state that "...this trend imposed by iWUE can be neutralized by increasing ca." Firstly, what trend do you mean? The one in ci/ca? Secondly, I am struggling with the logic of this sentence, since the principal effect of rising ca is stomatal closure, which increases iWUE. So how can ca counteract this at the same time? Doesn't that depend on how strong stomata respond to ca, as you have mentioned at the beginning of the section? This on the other hand is strongly controlled by the stomatal model used. The Ball-Berry model predicts a proportional decrease of gs with ca and a constant ci/ca. Please clarify this argument, in particular the role of ca for iWUE. I'm also wondering why the effect of mesophyll conductance is not discussed at this point, even though its importance is underlined in one of the studies you have cited (Seibt et al. 2008)? What would change if it was explicitly considered?

- 3.2.2 The idea that the stomatal slope may be too high for the site is interesting. Indeed a recent compilation of this parameter (Lin et al. 2015, Nature climate change) showed significantly lower values for coniferous evergreen forests than for other vegetation types (note that the study uses a slightly different model, and that the slopes cannot be compared 1:1, but they should vary in the same manner). One could cite this reference and point out that there is a biological explanation for why the slope should be lower for coniferous vegetation compared to other vegetation types. One could further explicitly mention that a lower stomatal slope would also give a lower stomatal conductance for a given An, and thus reduce the model-observation mismatch. Note that this would also affect Vcmax.

- Section 3.3 is very interesting, but I wonder if there is some more information on why one approach should be preferred over the other? Here you show that the limited N formulation is inferior to the others, which is nice, but is there also some biological evidence for this? What I mean is that the one reference you cite here (Zaehle et al., 2014) could be backed up by other (non-modeling) studies.

- Conclusions: You state that the isotope measurements suggest a lower gs than the flux tower measurements. I'm not sure if I agree with that, since you didn't derive gs directly from the eddy covariance measurements, but rather used the Ball-Berry model with an uncalibrated stomatal slope to model gs. So if your stomatal slope parameter is inappropriate for the vegetation at the site, then your gs will be as well, but that can't be directly related to the eddy covariance data.

- Figure 1: what do the lines prior to 1850 represent? Is it necessary to show them here?

- Figure 8: in Panel A it says fractionation in the heading but discrimination in the caption. Please stick to one. Why didn't you show iWUE here?

- Figure 9: could be helpful to add sub-headings on top of each row indicating the N-formulation used.

Technical corrections: - I suggest mentioning the FLUXNET ID of the site (US-NR1) - P.9 line 16: Max Planck Institute for Biogeochemistry - P9, line 18: remove brackets - P.5 line 13: remove brackets - Omit sentences like "the source code was modified..." - The horizontal lines of the error bars seem a bit overdimensioned - P. 21, line 19: "through", not "though"

---

## Author Comment (AC1) · 15 Jun 2016

Initial Response to Referee and Other Comments

An observational constraint on stomatal function in forests: evaluating coupled carbon and water vapor exchange with carbon isotopes in the Community Land Model (CLM 4.5)

Brett Raczka, Henrique F. Duarte, Charles D. Koven, Daniel Ricciuto, Peter E. Thornton, John C. Lin, David R. Bowling

Manuscript #: doi:10.5194/bg-2016-73, submitted Mar 22, 2016

Thank you to the referees, editor and Dr. Keeling for their time and thorough reviews. A combined response to these comments are below.

The most impactful comments by the reviewers in our opinion are related to the potential impact of mesophyll conductance upon the simulations. Please see our responses to the reviewer questions related to this topic. (Referee 1.4, 1.20, 1.23, 2.8).

Second in importance, we agree with the reviewer comments that a figure devoted to tracking carbon flow and distinguishing between the different model formulations would be useful. Please see the responses to reviewer questions related to this topic. (Referee 1.10, Ralph Keeling 2).

All suggested or proposed changes as described below are already implemented in a revised version of this manuscript. If given the opportunity, we plan to submit the revised manuscript to Biogeosciences.

On behalf of all authors,

Dr. Brett Raczka

brett.raczka@utah.edu

University of Utah
* * *
Response to comments of Referee 1:

Referee 1.1: The manuscript by Raczka et al. provides an extensive description of how carbon isotope discrimination is represented in the CLM land surface model and how it can be used as a constraint for evaluating model performance. The model evaluation conducted in this paper provides valuable insights into the representation of physiological processes in process-based models, including potential improvements with regard to model structure and parameterization. This information will be useful to the wider land surface modeling community.

Author: Thank You

Referee 1.2: The abstract could explicitly mention that three different N-limitation formulations were tested in the model. This information is worth to mention, but somewhat hidden in the abstract. E.g. it would make sense to shortly explain what the "alternative nitrogen limitation" formulation in line 29

actually means. To compensate for the additional number of words one could shorten the Vcmax calibration description or try to focus on a few key outcomes.

Author: We propose to clarify this by using the 'pre-photosynthetic' vs. 'post-photosynthetic' terminology used later in the manuscript.   We believe this would improve upon the 'alternative nitrogen formulation' description, but leaves the necessarily detailed description for the main body of the manuscript (too long for abstract).

Referee 1.3:  The original source of the Ball Berry model (Ball et al. 1987) should be acknowledged. Further, hs represents relative, and not specific humidity at the leaf surface. The definition of hs is unnecessary.

Author:  We intend to add this citation.  We will also correct $h_s$ to read: relative humidity.  Thanks for catching this.

Referee 1.4: ci is intercellular, not intracellular CO2 partial pressure, unless you consider mesophyll conductance, which seems not to be the case.

Author:  Thanks for catching this terminology mistake.  We will correct this, and we intend to add discussion that highlights that CLM ignores mesophyll conductance, and discuss the implications.

Referee 1.5: Equation 13: what does ET represent? In Table 1 it is listed as leaf transpiration, but that clearly doesn't make sense here. But I wonder if it is ecosystem transpiration or evapotranspiration?

Author:  We intend to change $E_T$ to ecosystem transpiration in Table 1.

Referee 1.6:  Table 1: please check the unit for iWUE, it shouldn't be gC gH2O-1. Add CO2 and O2 for the Michaelis Menten constants.

Author:  The units for iWUE should be changed to $\mu$mol C mol $H_2O^{-1}$ and we will add $CO_2$ and $O_2$ for the constants.

Referee 1.7:  In Equation 14, I assume you mean "An" rather than "A". Please clarify.

Author:  We mean "A" or gross assimilation as opposed to net assimilation "An" which includes a leaf respiration term.  To clarify we intend to define "A" more clearly in the methods and include a definition in Table 1 as well.

Referee 1.8: Please provide latin names for the dominant species at the site.

Author:   We will add the latin names.

Referee 1.9: Equation 8: please state where the 4.4 and 22.6 come from and which of those represents fractionation due to diffusion and Rubisco. I am not sure if this is clear to all readers.

Author:  We intend to make this clear in the text.  The values 4.4 and 22.6 represent the diffusional and enzymatic contributions to isotopic discrimination during photosynthesis.

Referee 1.10:  2.1.2: The comparison of different versions of how nitrogen limitation is implemented in the model and its implications is a very interesting aspect covered by the manuscript. Unfortunately, the

three different formulations tested (unlimited N, limited N, no downregulation discrimination) are described in a rather confuse way, and I doubt that it will comprehensible for all readers. I strongly encourage the authors to include the overview figure that they have shown in an earlier comment

Author:  We intend to add a new figure (similar to what was proposed and submitted to the interactive comments session).    It will track the flow from photosynthate to allocation of biomass.  It will also pictorially, show the role of the nitrogen cycle in the downregulation of photosynthesis.

Referee 1.11:  I recommend a better explanation of Equation 7. How is N-limitation determined? This is mentioned in the Figure caption, but one could also include this in the manuscript as well. Further, the terms "potential" and "actual photosynthesis" are mentioned on page 7, line 17f, but they haven't been defined before, and they aren't common terms either.   In the standard (= limited nitrogen) version, is photosynthesis first calculated without N-limitation, then N-limitation calculated according to Equation 7, and then the actual photosynthesis calculated by An*(1-fdreg)?

Author:  This will be better addressed by adding a new figure as proposed in the comment by Referee 1.10.  Also we will add two new equations that define how potential GPP is calculated from $A_n$, and that potential GPP is downscaled with $f_{dreg}$.   We left these out previously for conciseness.

Referee 1.12:  How is it possible that a reduction in An caused by N-limitation does not feedback on gs? This should be the case considering Eq. 4. The approach becomes clearer after reading section 3.3, but it would be helpful to explain it better at this point.

Author:  '$A_n$' (leaf-level photosynthesis) does not undergo a reduction from N-limitation, only GPP (ecosystem photosynthesis).   This will be much clearer with the addition of the new equations proposed within Referee 1.11.  In general, the fact that N-limitation does not feedback on $A_n$ and $g_s$, makes CLM a 'partially' coupled model.  We intend to add discussion to this effect.

Referee 1.13:   2.2 State here that NEE and other fluxes are observations based on the eddy covariance method. Please clarify here that the NEE partitioning was conducted using two different methods, and briefly mention their approach.

Author:  We will make these changes.

Referee 1.14:   P.10 line 27ff: that's a very detailed description which seems unnecessary to me. One could shorten this part or omit completely. - Same is true for the last sentence in 2.3 and the first sentences of 2.4., where many technical and CLM-specific details are mentioned that one may consider to omit, as they are of lesser interest to the wider community.

Author:  We intend to greatly simplify the explanation of the synthetic $CO_2$ and $\delta^{13}C$ time series in sections 2.3.1, 2.3.2 and move the details to the Methodological details section of the supplement.  This will also serve to limit the length of the document, a concern for some reviewers.

Referee 1.15: Figure 2: I think it would make more sense to show a mean annual course of the three variables rather than the complete time series. The way it is now makes it hard to see by how much GPP, ER, and LE differ from the observations on average.

Author:  Good idea.  We intend to update this figure to show the average seasonal cycle in fluxes.   We will move the original figure 2 to the supplement, not only to provide the length of the data record, but also to demonstrate transient behavior as revealed by the flux data (i.e. changes in productivity, or latent heat exchange by drier conditions etc.)

Referee 1.16:  An interesting aspect is the underestimation of WUE. Is this more related to evaporation or transpiration? In the latter case this would be strongly related to the stomatal slope parameter "m" in the Ball-Berry model (see later comment), but could have other reasons as well. One could shortly comment on this, up to the authors.

Author:  The manuscript does not address the accuracy of modeled WUE, given we don't have observations of transpiration and therefore no observations of WUE, only observations of latent heat flux, which the model simulates quite well after calibration (see Figure 2).

Referee 1.17: P.16 line 13: Please make sure that the iWUE trend reported in the studies cited here refer to the same time period. Over which timespan did the 15-20% increase in iWUE occur according to these studies?

Author:  We intend to clarify that this occurs over the time frame of 1960-2000.

Referee 1.18:  3.2.1 you state that ". . .this trend imposed by iWUE can be neutralized by increasing ca." Firstly, what trend do you mean? The one in ci/ca?

Author:  We mean the established relationship between iWUE and discrimination, that is, as iWUE increases, discrimination weakens (Saurer et al. 2004; equation 17).   We discuss this in the previous paragraph, and intend to edit the text to emphasize what relationship/trend we mean.

Referee 1.19:  Secondly, I am struggling with the logic of this sentence, since the principal effect of rising ca is stomatal closure, which increases iWUE. So how can ca counteract this at the same time? Doesn't that depend on how strong stomata respond to ca, as you have mentioned at the beginning of the section? This on the other hand is strongly controlled by the stomatal model used. The Ball-Berry model predicts a proportional decrease of gs with ca and a constant ci/ca. Please clarify this argument, in particular the role of ca for iWUE.

Author:  Equation 17 defines an inverse relationship between iWUE and ci*/ca (full derivation of this relationship can be found in supplement).  Equation 17 suggests that ci*/ca (discrimination) should decrease as a result of iWUE increasing (constant ca).   However,   if ca is also increasing at the same time this relationship between iWUE and discrimination can weaken.  Because iWUE should respond to an increase in ca through gs (as you have commented), this implies a weak stomatal response to ca in the model.

Referee 1.20:   I'm also wondering why the effect of mesophyll conductance is not discussed at this point, even though its importance is underlined in one of the studies you have cited (Seibt et al. 2008)? What would change if it was explicitly considered?

Author:  We intend to add discussion of the implications of ignoring mesophyll conductance upon our discrimination results.  In particular we intend to discuss the possibility that ignoring mesophyll conductance could have contributed to the overestimation of the simulated photosynthetic fractionation.  We also intend to add discussion that explains one of the key reasons that iWUE and discrimination can vary independently is because the model that Seibt et al. 2008 uses that relates iWUE and $\delta^{13}C$, includes mesophyll conductance.   They demonstrate that trends do emerge that are different from the linear model used by Saurer et al. (2004), consistent with our simulation results (increasing WUE and increasing discrimination).   This finding is largely coincidental, considering that CLM does not include mesophyll conductance, however, it is still important to show the model is not necessarily in conflict with observed trends.

Referee 1.21:  3.2.2 The idea that the stomatal slope may be too high for the site is interesting. Indeed a recent compilation of this parameter (Lin et al. 2015, Nature climate change) showed significantly lower values for coniferous evergreen forests than for other vegetation types (note that the study uses a slightly different model, and that the slopes cannot be compared 1:1, but they should vary in the same manner). One could cite this reference and point out that there is a biological explanation for why the slope should be lower for coniferous vegetation compared to other vegetation types. One could further explicitly mention that a lower stomatal slope would also give a lower stomatal conductance for a given An, and thus reduce the model-observation mismatch. Note that this would also affect Vcmax.

Author:  Thank you for bringing this to our attention.   The idea that the stomatal slope is too high leading to a stomatal conductance that is too high is also consistent with our simulated mismatch in discrimination (i.e. it is too high). We will add the Lin et al. 2015 paper to bolster this argument.

Referee 1.22: Section 3.3 is very interesting, but I wonder if there is some more information on why one approach should be preferred over the other? Here you show that the limited N formulation is inferior to the others, which is nice, but is there also some biological evidence for this? What I mean is that the one reference you cite here (Zaehle et al., 2014) could be backed up by other (non-modeling) studies.

Author:  Referee 2 offers DeKauwe et al., (2013) that provides site based observations within the FACE experiment at Duke and Oak Ridge, that indicate that fully coupled $A_n$-$g_s$ models tend to perform better in terms of GPP and WUE response to increased $CO_2$.  We intend to add discussion of how the partially-coupled version of CLM that we use in our manuscript are consistent with DeKauwe.

Referee 1.23:  Conclusions: You state that the isotope measurements suggest a lower gs than the flux tower measurements. I'm not sure if I agree with that, since you didn't derive gs directly from the eddy covariance measurements, but rather used the Ball-Berry model with an uncalibrated stomatal slope to model gs. So if your stomatal slope parameter is inappropriate for the vegetation at the site, then your gs will be as well, but that can't be directly related to the eddy covariance data.

Author: After calibration of Vcmax the simulated fluxes matched the flux tower observations much better (Figure 2), which makes our calibrated set of parameters consistent with the eddy covariance flux tower data, and biomass observations.   We think it is reasonable to suggest the stomatal slope is too high considering that other studies suggest the stomatal slope should be relatively low for coniferous evergreen species (Lin et al. 2015; Mao et al. 2016) (we intend to add the Lin et al. 2015), and that the

simulation is overestimating discrimination –consistent with a stomatal slope (stomatal conductance) that is too high.

However, we agree with the reviewer that because the stomatal slope parameterization was not taken directly from leaf-gas exchange measurements at the site, therein lies a possibility that calibrating the stomatal slope value to match isotopic discrimination could be in fact compensate for other parametric or structural errors within CLM.   We plan to discuss this possibility, and that we may be able to correct for bias in discrimination by including a representation of mesophyll conductance in CLM.

Referee 1.24:  Figure 1: what do the lines prior to 1850 represent? Is it necessary to show them here?

Author:  We changed the limits for the first column to start at 1850.

Referee 1.25:  Figure 8: in Panel A it says fractionation in the heading but discrimination in the caption. Please stick to one.

Author:  We will change this heading.

Referee 1.26:  I suggest mentioning the FLUXNET ID of the site (US-NR1) - P.9 line 16: Max Planck Institute for Biogeochemistry - P9, line 18: remove brackets - P.5 line 13: remove brackets - Omit sentences like "the source code was modified. . ." - The horizontal lines of the error bars seem a bit over-dimensioned - P. 21, line 19: "through", not "though"

Author:  We will make these changes.
* * *
Response to comments of Referee 2:

Referee 2.1:  I found the introduction very clear but I wonder if there is any other literature on how other models have used isotope data? I realize the authors suggest this is the first time it has been attempted in CLM and I realise this paper is primarily targeted at the CLM community, nevertheless I think my one concern would be the lack of literature in relation to other models and isotopes?

Author:   As this referee points out later in the review:  "… [suggests] cuts that could be made to the text which would make it more digestible.";  here is an instance where we felt we needed to be concise.  We do make references to previous isotope literature as was relevant to our work, for example Mao et al. (2016) and Aranibar et al. (2008).   Nevertheless, to address the reviewer's concern we will add a sentence that describes other isotope enabled land surface models.

Referee 2.2:  Equation 1: I don't think you mean Respd = dark respiration. Rdark is not the same as day respiration/respiration in the light. Suggest the use of Rday or Rd.

Author:  Correct.  The CLM literature (Oleson et al. 2013) uses the term '$R_d$' to describe this maintenance respiration term.  For this manuscript, we have purposely named this term '$Resp_d$' to prevent confusion with the isotope community convention of using 'R' to describe the ratio of $^{13}C$ and $^{12}C$ isotopes.   To

address the reviewer's concern we will refer to this term as just 'leaf respiration' in the text and Table (1).

Referee 2.3:   Equation 4: I'm pretty sure that "Bt" should be applied to your slope term "m", rather than the minimum stomatal conductance, b? Can you please check you have this correct?

Author:  This is correct as we have defined in Equation (4).   See Oleson et al. (2013).

Referee 2.4:  Line 23: "tree canopy" is this only true for trees, what happens with grasses in the model? If not, perhaps delete tree and leave just canopy.

Author:   We will replace "tree" with "vegetation" to avoid confusion.   In this manuscript CLM simulates the Niwot Ridge vegetation as a temperate evergreen needleleaf forest as already stated.   Grasses are not considered in this manuscript, but CLM is capable of simulating grasses.

Referee 2.5:  Equation 8 & 9: it would be helpful to the reader to explain where the numbers 4.4, 22.6 and 1000 come from, or what conversions they apply to.

Author:  We intend to edit the text to make a clearer linkage between the numbers and the fractionation mechanism they represent.

Referee 2.6:  Century model (line 26/27) should have a reference.

Author:  We will add a reference (Parton 1988).

Referee 2.7:  I'm not sure what the length of the paper was but the results/discussion text did feel very long? Similarly the conclusions runs to nearly two pages. This seems excessive to me. I'm fairly confident there are cuts that could be made to the text which would make it more digestible to the reader. I certainly found myself losing track during my reading and I think this is the key area which requires editing during revision.

Author:  We found this suggestion difficult to address given its generality.  However, where Reviewer 1 made specific suggestions of cuts within the Methods (2.3.1, 2.3.2) we intend to cut roughly 20 lines of text from the Methods section. We also intend to cut ~ 10 lines of text within the conclusions.  The revised manuscript will remain the same length (26 pages) even with significant discussion added.

Referee 2.8:  The authors note: "the overestimation of discrimination may suggest the stomatal slope in the Ball-Berry model (m=9 in Eq. 4) used for these simulations was too high." While it is may be true that the slope parameter is poorly informed by site data, the logic of this conclusion in itself may not be valid. Isotopic measurements *should* give lower slope values than those one would infer via leaf gas exchange data (i.e. the data used to inform the Ball-Berry model). This is because leaf gas exchange measures the resistance from the intercellular spaces (Ci), whereas isotopes measures the resistance from the chloroplast (Cc). I see no mention of this in the text and caution against the authors potential drawing the wrong conclusion from the model-data discrepancy.

Author:  We intend to address mesophyll conductance specifically as discussed above.  First, our finding that the stomatal slope parameter value is likely too large is a reasonable conclusion for 2 reasons:  1) A lower stomatal slope value is consistent with both model results (Mao et al. 2016) and leaf-gas exchange measurements (Lin et al. 2015).   Discussion of the Lin et al paper will be added.  Second, a lower

stomatal slope value will lead to a lower stomatal conductance which will help reduce the overestimation of the modeled isotopic discrimination (Figure 7).

With that being said, we will add in the discussion the possibility that this result may, in part, come from the simplified approach of CLM 4.5, that does not specifically include mesophyll conductance and assumes intercellular CO2 = intracellular CO2. Therefore, we will add this caveat, that the need to reduce the stomatal slope, may be the result of missing mechanisms governing mesophyll conductance within CLM. Given the potential importance to the paper, we will include this possibility in the abstract.

Referee 2.9: In discussing the "limited nitrogen formulation", the authors note: "In general, there were no categorical differences in behavior between these two classes of models during CO2 manipulation experiments held at Duke forest and ORNL (Zaehle et al., 2014). CLM 4.0 was one of the few models in that study to consistently underestimate the NPP response to an increase of atmospheric CO2 due to nitrogen limitation, however this finding was attributed to a lower initial supply of nitrogen." This is not strictly true. As part of the same model-data inter-comparison of the models to the data at the two FACE sites, De Kauwe et al. (2013, Global Change Biology), found no support for the implementation whereby assimilation is limited by nitrogen availability, but not stomatal conductance. They concluded: "Stomatal conductance data from both sites were used to test modelled leaf-level responses. The simple stomatal conductance model (Eq. 1) fitted the data well (Fig. 6), supporting the assumption of coupling between assimilation and stomatal conductance. Importantly, at the ORNL site, N content of the foliage declined strongly over the course of the experiment (Norby et al., 2010), but neither the slope of the stomatal model, nor the response of A/gs to CO2, was altered by this decline (Fig. 6b). These data indicate that the coupling between stomatal conductance and assimilation is not affected by N-limitation (Fig. 6b). The data therefore tend to support coupled models over uncoupled, or partially coupled, models such as DAYCENT and CLM4." Furthermore, I would question if there is any evidence that plants follow the "limited nitrogen formulation"?

Author: Thank you for bringing De Kauwe et al. 2013 to our attention. First, we may be talking about two different sub-groupings of models: in our manuscript we are comparing pre-photosynthetic (foliar nitrogen) and post-photosynthetic nitrogen limitation models. Almost all of these models regardless of pre/post photosynthetic sub-grouping contain stomatal-photosynthetic coupling though Ball-Berry type assumptions in the stomatal conductance model.

It is true that CLM4 and CLM4.5 in the default model (post-photosynthetic model formulation) is only 'partially' coupled in terms of photosynthetic-stomatal conductance, however the unlimited nitrogen formulation (pre-photosynthetic) in our manuscript is 'fully' coupled ($A_n$ is consistent and solved simultaneously with $g_s$). Therefore, our simulations were consistent with De Kauwe et al. 2013 in that fully-coupled models matched the observations the best. We will add this to the discussion.

This progressive de-coupling between $A_n$ and $g_s$ for our default CLM 4.5 version also explains the difference in transient behavior between $g_s$ and $A_n$ and iWUE as shown in Figure (6). We will add this to discussion.

This referee makes another comment that seems to be referring to a 3rd sub-grouping of model – models that do not consider nitrogen limitation at all – similar to the simple stomatal-assimilation model in De Kauwe. What role does nitrogen limitation play (if any) in assimilation and stomatal behavior? We are not sure how much our manuscript can inform this question. Clearly, the default version of CLM 4.5

(post-photosynthetic formulation) is strongly influenced by the nitrogen cycle, whereas our pre-photosynthetic formulation is less strictly linked to the nitrogen cycle (Vcmax was calibrated to match eddy covariance flux observations, not according to nitrogen constraints).   However, even for the pre-photosynthetic formulation it is plausible that leaf nitrogen content plays a role in the Vcmax value. Therefore within the limitations of this manuscript we don't think we can comment on the significance of nitrogen limitation on ecosystem behavior, but only that if nitrogen limitation is implemented, it should occur pre-photosynthetically.

Referee 2.10:  Figure 2. I realize that a strength of this paper is the long time series; however, showing ~15 years of data like that isn't particularly instructive. It is hard to distinguish the model-obs differences. Perhaps average a day/week or monthly climatology across years would more clearly show differences. This figure could also be kept, perhaps one could go to the supplementary.

Author:   We intend to edit figure 2 to provide a seasonally averaged flux behavior across all years, to better illustrated model-observation differences, and calibrated/uncalibrated model differences.   We intend to move the original figure 2 to the supplement, not only to provide the length of the data record, but also to demonstrate any transient behavior as revealed by the flux data (i.e. changes in productivity, or latent heat exchange by drier conditions etc.)

Referee 2.11:  Figure 8e. I find it hard to believe that there is no reduction in the soil moisture availability factor during the whole of the summer? This seems unlikely to me? Could this please be checked?

Author:  We thought the same thing. The modeled soil moisture tends to compare favorably to the observed soil moisture as measured at multiple depths both in terms of the magnitude and seasonal trends.   In general, the modeled soil moisture tends to simulate slightly wetter conditions as we already point out in the discussion, but per communication with site PI's, the accuracy of the soil moisture sensors is questionable.   Therefore we did not attempt to calibrate the hydrology model to best match the soil moisture sensors.  We hypothesize that the modeled soil moisture is too wet at depth, thereby leading to little change in BTRAN (soil moisture stress parameter).  We chose to limit this discussion in the text to keep the manuscript length in check, a concern for this reviewer.

Referee 2.12:  Figure 9. I would suggest the symbol sizes could be reduced, they seem a little large for the figure panels.

Author:  We will reduce the symbol size.
* * *
Response to comments by Ralph Keeling:

Ralph Keeling 1:  Overall, very nice paper, which I found quite educational. I'm passing on here a few comments that I jotted down as I read through the paper.

Author:  Thank you.

Ralph Keeling 2:   Perhaps it would be possible (?) to add a figure which diagrams the carbon flows from the atmosphere, through stomata to substrate formation for each of the three formulations. I'm imagining that the diagram would have arrows for each of these quantities: An, GPP, CFavailable_alloc, CF_alloc, CF_GPPpot, etc. Or maybe one figure would suffice, assuming the knobs to switch between formulations is clear enough.

Author:  We intend to add a new figure (Figure 1), similar to what we proposed and posted in the discussion forum, which explicitly tracks the carbon flows through the 2 main nitrogen sub-models used in this study, and illustrates how the nitrogen limitation model interacts with these carbon flows from substrate to biomass.  We also intend to add two new equations which explicitly show the linkage between $A_n$ and $CF_{GPPpot}$ and between $CF_{GPPpot}$ and GPP.  This new figure, and the new equations, combined with the existing Table 2 should provide a better overview of the sub-models that complements the text to enhance reader understanding.

Ralph Keeling 3:  ….where is the carbon that is fixed but not allocated ending up? Is it respired? If so, does this respiration return back from the stomata or return through some other pathway?

Author:  For the limited nitrogen sub-model (post-photosynthetic limitation), the carbon that is fixed but not allocated, is removed from the system (does not show up as a respired flux).  This is arguably a weakness in this version of the model:  the downscaled assimilated flux is not consistent with the carboxylation rate ($A_n$) and stomatal conductance ($g_s$) that created the pre-downscaled flux, and is why this version of CLM is considered to be 'partially' coupled, which we intend to add to the discussion.  The unlimited nitrogen sub-model (pre-photosynthetic limitation) is not subject to this apparent inconsistency and is 'fully' coupled.  We intend to add a new Figure (1), similar to what was posted in the discussion forum, that shows a valve for this downscaling, and no respired flux.  We also intend to add clarification that states that this excess carbon is lost to the system, and does not show up as a respired flux.

Ralph Keeling 4:  Page 6, line 28: I'm missing how An is related to terms in Eq. (6). It would very much help to include an algebraic expression for this.

Author: We will add a new equation that relates An to potential GPP term $CF_{GPPot}$.

Ralph Keeling 5: Page 6, lines 29-30. From the wording it sounds like maintenance respiration is partly double counted.

Author: The $CF_{GPP,mr}$ term comes directly from the carbon pool from photosynthesis (photosynthate).  When there is no photosynthesis the model calls on a storage carbon pool to meet this demand:  $CF_{GPP,xs}$.  The maintenance respiration is coming from the photosynthate pool, and when photosynthate is low or zero, is supplemented by the maintenance respiration storage pool.   We will clarify this.

Ralph Keeling 6:  Page 7, line 11. What does the subscript psn signify? Perhaps could be omitted?

Author:  $P_{sn}$ stands for photosynthetic fractionation.   This is implied in the context of the manuscript so it will be removed throughout.

Ralph Keeling 7: Page 7, line 15: This formula suggests that An is not equal to the flux through stomata. So what is An equal to? Is it the same as potential photosynthesis? If so, needs stating. See earlier comment also.

Author: $A_n$, as defined in equation (1) is the (potential) leaf-level net assimilation rate which is used to calculate potential photosynthesis ($CF_{GPPpot}$).   We will specify that $A_n$ is the leaf-level net carbon assimilation.  We also add a new equation that connects $A_n$ with $CF_{GPPpot}$ based on a previous comment making it clear that $A_n$ is the assimilation rate that is used to calculate potential GPP.

Ralph Keeling 8:  Page 7, lines 22-23: This sentence is a bit ambiguous. Are both given in Eq. 9, or just one. If not both, then how is nitrogen limitation incorporated? Reading below, I see this is probably related to control of Vcmax. If so, this need stating more clearly earlier.

Author:  Both formulations follow Equation (9).  We intend to make this clearer in the text:   "The unlimited nitrogen formulation also follows equation (9), however the vegetation is allowed to have unlimited access to nitrogen."

We intend to add discussion in the next paragraph of the manuscript that gives a thorough explanation how we used Vcmax in order to take into account for nitrogen limitation, even when the nitrogen downregulation factor is not used.

Ralph Keeling 9: Page 8, line 22. I'm missing an expression for how delta_GPP is calculated from alpha_psn. (Okay, I know enough to work this out for myself, but I'm not sure you should assume all readers would).

Author:  The fractionation factor $\alpha$ is defined in the beginning of section 2.1.2, stating that $\alpha = R_a/R_{GPP}$. This relates $\alpha$ to $R_{GPP}$. One can then use equation (10) to get from $R_{GPP}$ to $\delta_{GPP}$.

Ralph Keeling 10:  Page 8, line 28. It would seem important to clarify what is meant here by GPP. Which of these is it: An, CFavailable_alloc, CFGPP_pot, etc. ?

Author: This GPP is the final downscaled GPP or actual GPP (ecosystem photosynthesis). We will add a new equation that defines how GPP is downscaled through $f_{dreg}$, and this downscaled, or 'actual' GPP is what is used in the definition for WUE.

---

## Author Response (AR1)

Response to Referee and Other Comments

An observational constraint on stomatal function in forests: evaluating coupled carbon and water vapor exchange with carbon isotopes in the Community Land Model (CLM 4.5)

Brett Raczka, Henrique F. Duarte, Charles D. Koven, Daniel Ricciuto, Peter E. Thornton, John C. Lin, David R. Bowling
Manuscript #: doi:10.5194/bg-2016-73, submitted Mar 22, 2016

Thank you to the referees, editor and Dr. Keeling for their time and extremely thorough reviews. Their comments have greatly improved the manuscript. A combined response to these comments are below. Line numbers in the author response refer to the revised manuscript (not the track changes version). Also note that the referee comments refer to pages, line numbers and figures from the original manuscript that no longer match with the revised manuscript. Each comment is numbered in the order presented by each referee.

The most important addition to the revised manuscript reflects the potential impact of mesophyll conductance upon the simulations – commented upon by both reviewers. We have addressed this in the manuscript abstract (page 2, lines 13-14) discussion (page 19, lines 3-17) and conclusions (page 25, lines 26-31). Also please see the responses to the reviewer questions related to mesophyll conductance. (Referee 1.4, 1.20, 1.23, 2.8).

Secondly, we have included a new figure (Figure 1) that was suggested by the reviewers and Dr. Keeling to help clarify the carbon flow within the CLM model, including the impact of nitrogen limitation within the two major formulations of CLM in our analysis. We refer to it within the manuscript (page 8, lines 13-14 and page 8, lines 25-26). Also please see the responses to reviewer questions related to this topic. (Referee 1.10, Ralph Keeling 2)

On behalf of all authors,

Dr. Brett Raczka brett.raczka@utah.edu

University of Utah
* * *
Response to comments of editor:

Editor 1: Many thanks for your replies to the reviewer comments. Please provide a thoroughly revised version, addressing each of the comments and ensuring that your responses are reflected by changes in the manuscript.

Author: We have provided a fully revised manuscript based upon the reviewers, editor and Dr. Keeling's comments. We also have provided a track changes version of this revised manuscript.

Editor 2: I don't think that the terminology of 'pre-photosynthetic' vs. 'post-photosynthetic' N limitation is useful. Rather I would recommend to state that either photosynthesis or growth is assumed to acclimate to N availability.

Author: We believe the terminology of 'pre-photosynthetic' vs. 'post-photosynthetic' N limitation is the most straightforward way to define the two major model formulations. Admittedly, this description makes the most sense from a modeling point of view and not necessarily from an ecological perspective, however, we devote considerable discussion describing the model formulations from an ecological perspective in Section 2.1.2. We think the editor's suggestion to define the 'post-photosynthetic' formulations as 'acclimating to N availability' is a useful way to distinguish between the two formulations and we have added discussion (page 8, lines 26-29) to reflect this. We caution that to equate the 'post-photosynthetic' formulation as the only formulation that 'acclimates to N availability' is not strictly true – even the 'pre-photosynthetic' formulation which uses a calibrated Vcmax likely reflects, in part, the influence of nitrogen availability.

Editor 3: The use of gross photosynthesis is not really helpful when defining intrinsic water-use efficiency, because as a consequence of the varying temperature response of A and leaf dark respiration iWUE will vary with temperature, which it should not. I do not see a justification for not using An.

Author: We anticipated that the % change of iWUE whether using net assimilation or gross assimilation rate would remain relatively unchanged given that the modeled climate (and therefore temperature) remained generally constant during the dynamic portion of the simulation. Nevertheless, we re-made Figure 7 using net assimilation instead of gross assimilation and found that the iWUE increased slightly more, from 15% to 20% from 1960-2000. We have updated Figure 7 and included discussion on page 16, lines 18-20 to reflect this change. We have also edited equation 7 and Table 2 to reflect this change in the definition of iWUE.

Editor 4: Please provide a reference for the discrimination values.

Author: If we understand the comment correctly, you mean the observed photosynthetic discrimination that we describe in Section 3.2.2, Figure 8 and Figure 10. In all these locations we reference Bowling et al. (2014) as the source for the photosynthetic discrimination values.

Editor 5: Your response to 1.12 is unsatisfying: How can GPP at the ecosystem level not be dependent on An on the leaf level. This makes absolutely no sense at all.

Author: Agreed, this is strange, but it is a result of the CLM model structure and does not reflect the expected ecological linkage. We have added several equations to the manuscript to make this clearer (Equations 6 and 7), not only for this question, but based on reviewer comment 1.11 (see below). The potential GPP is calculated from the potential leaf-level photosynthesis (Equation 6). Next, the potential GPP is downscaled through nitrogen availability (equation 9). This downscaling of potential GPP does not feed back to the leaf-level equations of $A_n$ and gs (equations 1 and 4), which is why CLM is considered a 'partially' coupled model. We have added discussion of this on page 16, lines 26-32, and this also serves as a response to comment 2.9 and Keeling 3.

The editor is correct that from an overall ecological perspective the actual (downscaled) GPP must result from an actual (downscaled) leaf level photosynthesis. This is calculated from CLM as: GPP/LAI. However, this downscaled leaf level photosynthesis is distinct from the modeled definition of $A_n$ in

Equation 1, and that is what we are referring to in our response to reviewer 1.12.  We have added this caveat in the response to reviewer 1.12.

Editor 6:  Your response to 1.16.  It should be possible to calculate the fraction of LE which is driven by transpiration and therefore identify whether the canopy or soil fluxes contribute to the WUE bias.

Author:  We interpreted comment 1.16 as referring to the observed intrinsic water use efficiency,  data for which are not available.   We assume from the reviewer comment that the WUE bias that is referred to is the underestimation of the increase in simulated iWUE as described in Figure 7 panel F and discussed in section 3.2.1.   Now that we have updated iWUE to be the ratio of the *net* assimilation to stomatal conductance and not *gross* assimilation to stomatal conductance (see editor comment 3) the simulated increase in iWUE (20% -limited nitrogen, 10% unlimited nitrogen) now spans the range from other literature (15-20%), making the simulation and observations indistinguishable.   We have updated page 16, lines 16-20 to reflect this change.   In response to the reviewer's comment, we did examine the simulated mean fraction of transpired latent heat, which was 70-75% during the summer months.  We don't feel it is within the scope of the present manuscript to add this now that the WUE bias is gone, however, we can add it the supplemental material at the editor's discretion.
* * *
Response to comments of Referee 1:

Referee 1.1: The manuscript by Raczka et al. provides an extensive description of how carbon isotope discrimination is represented in the CLM land surface model and how it can be used as a constraint for evaluating model performance. The model evaluation conducted in this paper provides valuable insights into the representation of physiological processes in process-based models, including potential improvements with regard to model structure and parameterization. This information will be useful to the wider land surface modeling community.

Author:   Thank You

Referee 1.2: The abstract could explicitly mention that three different N-limitation formulations were tested in the model. This information is worth to mention, but somewhat hidden in the abstract. E.g. it would make sense to shortly explain what the "alternative nitrogen limitation" formulation in line 29 actually means. To compensate for the additional number of words one could shorten the Vcmax calibration description or try to focus on a few key outcomes.

Author: We have clarified this by using the pre-photosynthetic vs. post-photosynthetic terminology within the abstract, which was used later in the manuscript (page 1-2, line 27-29 and 1).   We believe this improves upon the 'alternative nitrogen formulation' description, but leaves a necessarily detailed description for the main body of the manuscript (too long for abstract).

Referee 1.3:  The original source of the Ball Berry model (Ball et al. 1987) should be acknowledged. Further, hs represents relative, and not specific humidity at the leaf surface. The definition of hs is unnecessary.

Author:  Citation added.  Corrected $h_s$ to relative humidity.  Thanks for catching this.

Referee 1.4: ci is intercellular, not intracellular CO2 partial pressure, unless you consider mesophyll conductance, which seems not to be the case.

Author: Thanks for catching this terminology mistake. This has been corrected, and there now is a paragraph in the discussion (page 19 lines 3-17) highlighting that CLM ignores mesophyll conductance, and discusses the implications.

Referee 1.5: Equation 13: what does ET represent? In Table 1 it is listed as leaf transpiration, but that clearly doesn't make sense here. But I wonder if it is ecosystem transpiration or evapotranspiration?

Author: $E_T$ was changed to ecosystem transpiration in Table 1.

Referee 1.6: Table 1: please check the unit for iWUE, it shouldn't be gC gH2O-1. Add CO2 and O2 for the Michaelis Menten constants.

Author: The units for iWUE were changed to $\mu mol\ C\ mol\ H_2O^{-1}$ and we added $CO_2$ and $O_2$ for the constants.

Referee 1.7: In Equation 14, I assume you mean "An" rather than "A". Please clarify.

Author: We have changed Equation 14 to read An and have changed panel F Figure 7 (iWUE) to reflect this change.

Referee 1.8: Please provide latin names for the dominant species at the site.

Author: We added latin names on page 9, lines 24-25.

Referee 1.9: Equation 8: please state where the 4.4 and 22.6 come from and which of those represents fractionation due to diffusion and Rubisco. I am not sure if this is clear to all readers.

Author: This is now made clear in the text, (page 7, lines 21-23) that 4.4 and 22.6 represent the diffusional and enzymatic contributions to isotopic discrimination during photosynthesis.

Referee 1.10: 2.1.2: The comparison of different versions of how nitrogen limitation is implemented in the model and its implications is a very interesting aspect covered by the manuscript. Unfortunately, the three different formulations tested (unlimited N, limited N, no downregulation discrimination) are described in a rather confuse way, and I doubt that it will comprehensible for all readers. I strongly encourage the authors to include the overview figure that they have shown in an earlier comment

Author: We have added a new Figure 1 to help make this clear. We also refer to this figure on page 8, line 13-14 and page 8, line 25-26.

Referee 1.11: I recommend a better explanation of Equation 7. How is N-limitation determined? This is mentioned in the Figure caption, but one could also include this in the manuscript as well. Further, the terms "potential" and "actual photosynthesis" are mentioned on page 7, line 17f, but they haven't been defined before, and they aren't common terms either. In the standard (= limited nitrogen) version, is photosynthesis first calculated without N-limitation, then N-limitation calculated according to Equation 7, and then the actual photosynthesis calculated by An*(1-fdreg)?

Author: This is now better addressed by adding Figure 1 and associated text (page 8, line 13-14 and page 8, line 25-26), and by adding equations (6 and 9) that define how potential GPP is calculated from $A_n$, and that potential GPP is downscaled with $f_{dreg}$.

Referee 1.12: How is it possible that a reduction in An caused by N-limitation does not feedback on gs? This should be the case considering Eq. 4. The approach becomes clearer after reading section 3.3, but it would be helpful to explain it better at this point.

Author: '$A_n$' (leaf-level photosynthesis) does not undergo a reduction from N-limitation, only GPP (ecosystem photosynthesis). This should be much clearer now with the addition of Eqs. 6 and 9 as discussed previously. In general, the fact that N-limitation does not feedback on $A_n$ and $g_s$, makes CLM a 'partially' coupled model. We added discussion to reflect this on page 16, line 26-32.

As pointed out by the editor (editor comment 5) to our initial response to comment 1.12, the $A_n$ we are referring to is from Equation 6, which technically is the 'potential' leaf level photosynthesis, and not the actual leaf level photosynthesis (after N downscaling).

Referee 1.13: 2.2 State here that NEE and other fluxes are observations based on the eddy covariance method. Please clarify here that the NEE partitioning was conducted using two different methods, and briefly mention their approach.

Author: We implemented these changes on page 10, lines 3-7.

Referee 1.14: P.10 line 27ff: that's a very detailed description which seems unnecessary to me. One could shorten this part or omit completely. - Same is true for the last sentence in 2.3 and the first sentences of 2.4., where many technical and CLM-specific details are mentioned that one may consider to omit, as they are of lesser interest to the wider community.

Author: We have simplified the explanation of the synthetic $CO_2$ and $\delta^{13}C$ time series in sections 2.3.1, 2.3.2 and moved the details to the Methodological details section of the supplement.

Referee 1.15: Figure 2: I think it would make more sense to show a mean annual course of the three variables rather than the complete time series. The way it is now makes it hard to see by how much GPP, ER, and LE differ from the observations on average.

Author: Good idea. Figure 3 has been updated to show the average seasonal cycle in fluxes. We moved the original figure 2 to the supplement (Figure S1), not only to provide the length of the data record, but also to demonstrate transient behavior as revealed by the flux data (i.e. changes in productivity, or latent heat exchange by drier conditions etc.)

Referee 1.16: An interesting aspect is the underestimation of WUE. Is this more related to evaporation or transpiration? In the latter case this would be strongly related to the stomatal slope parameter "m" in the Ball-Berry model (see later comment), but could have other reasons as well. One could shortly comment on this, up to the authors.

Author: The manuscript does not address the accuracy of modeled WUE, given we don't have observations of transpiration and therefore no observations of WUE, only observations of latent heat flux, which the model simulates quite well after calibration (see Figure 3).

Now that we have updated iWUE to be the ratio of the net assimilation to stomatal conductance and not gross assimilation to stomatal conductance (see editor comment 3) the simulated increase in iWUE (20% -limited nitrogen, 10% unlimited nitrogen) now spans the range from other literature (15-20%), making the simulation and observations indistinguishable.  We have updated page 16, lines 16-20 to reflect this change.   Diagnosing how much of the latent heat comes from transpiration is now, in our opinion, outside the scope of the manuscript, but we could add this to the supplement at the editor's discretion.

Referee 1.17: P.16 line 13: Please make sure that the iWUE trend reported in the studies cited here refer to the same time period. Over which timespan did the 15-20% increase in iWUE occur according to these studies?

Author:  We clarified the text to state this change occurred within the time frame of 1960-2000.  (Page 16, Lines 16-18)

Referee 1.18:  3.2.1 you state that ". . .this trend imposed by iWUE can be neutralized by increasing ca."  Firstly, what trend do you mean? The one in ci/ca?

Author:  We mean the established relationship between iWUE and discrimination, that is, as iWUE increases, discrimination weakens (Saurer et al. 2004; equation 19).   We discuss this in the previous paragraph (page 17, lines 12-29), and edited the text to emphasize what relationship/trend we mean (page 17, lines 30-31).

Referee 1.19:  Secondly, I am struggling with the logic of this sentence, since the principal effect of rising ca is stomatal closure, which increases iWUE. So how can ca counteract this at the same time? Doesn't that depend on how strong stomata respond to ca, as you have mentioned at the beginning of the section? This on the other hand is strongly controlled by the stomatal model used. The Ball-Berry model predicts a proportional decrease of gs with ca and a constant ci/ca. Please clarify this argument, in particular the role of ca for iWUE.

Author:  Equation 19 defines an inverse relationship between iWUE and ci*/ca (full derivation of this relationship can be found in the supplement).  Equation 19 suggests that ci*/ca (discrimination) should decrease as a result of iWUE increasing (constant ca).   However,   if ca is also increasing at the same time this relationship between iWUE and discrimination can weaken.  Because iWUE should respond to an increase in ca through gs (as you have commented), this implies a weak stomatal response to ca in the model.

Referee 1.20:   I'm also wondering why the effect of mesophyll conductance is not discussed at this point, even though its importance is underlined in one of the studies you have cited (Seibt et al. 2008)? What would change if it was explicitly considered?

Author: Please see added discussion of implications of ignoring mesophyll conductance on page 19, lines 3-17).  We also added discussion (page 17: lines 12-21) that explains one of the key reasons that iWUE and discrimination can vary independently is because the model that Seibt et al. 2008 uses that relates iWUE and $\delta^{13}C$, includes mesophyll conductance.   They demonstrate that trends do emerge that are different from the linear model used by Saurer et al. (2004), consistent with our simulation results (increasing WUE and increasing discrimination). This finding is largely coincidental, considering that CLM does not include mesophyll conductance, however, it is still important to show the model is not necessarily in conflict with observed trends.

Referee 1.21: 3.2.2 The idea that the stomatal slope may be too high for the site is interesting. Indeed a recent compilation of this parameter (Lin et al. 2015, Nature climate change) showed significantly lower values for coniferous evergreen forests than for other vegetation types (note that the study uses a slightly different model, and that the slopes cannot be compared 1:1, but they should vary in the same manner). One could cite this reference and point out that there is a biological explanation for why the slope should be lower for coniferous vegetation compared to other vegetation types. One could further explicitly mention that a lower stomatal slope would also give a lower stomatal conductance for a given An, and thus reduce the model-observation mismatch. Note that this would also affect Vcmax.

Author: Thank you for bringing this to our attention. The idea that the stomatal slope is too high leading to a stomatal conductance that is too high is also consistent with our simulated mismatch in discrimination (i.e. it is too high). We have added the Lin et al. 2015 paper to page 18-19, lines 27-29,1-2.

Referee 1.22: Section 3.3 is very interesting, but I wonder if there is some more information on why one approach should be preferred over the other? Here you show that the limited N formulation is inferior to the others, which is nice, but is there also some biological evidence for this? What I mean is that the one reference you cite here (Zaehle et al., 2014) could be backed up by other (non-modeling) studies.

Author: Referee 2 offers DeKauwe et al., (2013) that provides site based observations within the FACE experiment at Duke and Oak Ridge, that indicate that fully coupled $A_n$-$g_s$ models tend to perform better in terms of GPP and WUE response to increased $CO_2$. We have included discussion of this on pages: 22-23 lines 30-32, 1-3.

Referee 1.23: Conclusions: You state that the isotope measurements suggest a lower gs than the flux tower measurements. I'm not sure if I agree with that, since you didn't derive gs directly from the eddy covariance measurements, but rather used the Ball-Berry model with an uncalibrated stomatal slope to model gs. So if your stomatal slope parameter is inappropriate for the vegetation at the site, then your gs will be as well, but that can't be directly related to the eddy covariance data.

Author: After calibration of Vcmax the simulated fluxes matched the flux tower observations much better (Figure 3), which makes our calibrated set of parameters consistent with the eddy covariance flux tower data, and biomass observations. We think it is reasonable to suggest the stomatal slope is too high considering that other studies suggest the stomatal slope should be relatively low for coniferous evergreen species (Lin et al. 2015; Mao et al. 2016) (which we add discussion of page 18-19, lines: 23-29, 1-2), and that the simulation is overestimating discrimination –consistent with a stomatal slope (stomatal conductance) that is too high.
    However, we agree with the reviewer that because the stomatal slope parameterization was not taken directly from leaf-gas exchange measurements at the site, therein lies a possibility that calibrating the stomatal slope value to match isotopic discrimination could be in fact compensate for other parametric or structural errors within CLM. We discuss this possibility on page 19, lines 3-17 where we may be able to correct for bias in discrimination by including a representation of mesophyll conductance in CLM.

Referee 1.24:  Figure 1: what do the lines prior to 1850 represent? Is it necessary to show them here?

Author:  We changed the limits for the first column to start at 1850 in what is now Figure (2).

Referee 1.25:  Figure 8: in Panel A it says fractionation in the heading but discrimination in the caption. Please stick to one.

Author:  We changed heading to discrimination in what is now Figure (9).

Referee 1.26:  I suggest mentioning the FLUXNET ID of the site (US-NR1) - P.9 line 16: Max Planck Institute for Biogeochemistry - P9, line 18: remove brackets - P.5 line 13: remove brackets - Omit sentences like "the source code was modified. . ." - The horizontal lines of the error bars seem a bit over-dimensioned - P. 21, line 19: "through", not "though"

Author:  All changes were made.
* * *
Response to comments of Referee 2:

Referee 2.1:  I found the introduction very clear but I wonder if there is any other literature on how other models have used isotope data? I realize the authors suggest this is the first time it has been attempted in CLM and I realise this paper is primarily targeted at the CLM community, nevertheless I think my one concern would be the lack of literature in relation to other models and isotopes?

Author:   As this referee points out later in the review:  "… [suggests] cuts that could be made to the text which would make it more digestible.";  here is an instance where we felt we needed to be concise.  We do make references to previous isotope literature as was relevant to our work, for example Mao et al. (2016) Page 18, Line: 23 and Aranibar et al. (2008) Page 18, Line: 26.   Nevertheless, to address the reviewer's concern we have added a sentence that includes a list of isotope enabled land surface models.  Lines: page 4, lines 2-6.

Referee 2.2:   Equation 1: I don't think you mean Respd = dark respiration. Rdark is not the same as day respiration/respiration in the light. Suggest the use of Rday or Rd.

Author:  Correct.  The CLM literature (Oleson et al. 2013) uses the term '$R_d$' to describe this respiration term.  For this manuscript, we have purposely named this term 'Resp$_d$' to prevent confusion with the isotope community convention of using 'R' to describe the ratio of $^{13}C$ and $^{12}C$ isotopes.   To address the reviewer's concern we refer to this term as just 'leaf respiration' in the text and Table (1).

Referee 2.3:   Equation 4: I'm pretty sure that "Bt" should be applied to your slope term "m", rather than the minimum stomatal conductance, b? Can you please check you have this correct?

Author:  This is correct as we have defined in Equation (4).   See Oleson et al. (2013), page 183.

Referee 2.4:  Line 23: "tree canopy" is this only true for trees, what happens with grasses in the model? If not, perhaps delete tree and leave just canopy.

Author:  We replaced "tree" with "vegetation" to avoid confusion on page 6, line 25-26.   In this manuscript CLM simulates the Niwot Ridge vegetation as a temperate evergreen needleleaf forest as stated in page 12, line 11-14.   Grasses are not considered in this manuscript, but CLM is capable of simulating grasses.

Referee 2.5:  Equation 8 & 9: it would be helpful to the reader to explain where the numbers 4.4, 22.6 and 1000 come from, or what conversions they apply to.

Author:  We edited the text (page 7, lines 21-23) to make a clearer linkage between the numbers and the fractionation mechanism they represent.

Referee 2.6:  Century model (line 26/27) should have a reference.

Author:  We added a reference to page 12 line 3-5.

Referee 2.7:  I'm not sure what the length of the paper was but the results/discussion text did feel very long? Similarly the conclusions runs to nearly two pages. This seems excessive to me. I'm fairly confident there are cuts that could be made to the text which would make it more digestible to the reader. I certainly found myself losing track during my reading and I think this is the key area which requires editing during revision.

Author:  We found this suggestion difficult to address given its generality.  However, where Reviewer 1 made specific suggestions of cuts within the Methods (2.3.1, 2.3.2) we cut roughly 20 lines of text from the Methods section. We also cut ~ 10 lines of text within the conclusions.  The revised manuscript has remained the same length (26 pages) even with discussion added in response to other comments.

Referee 2.8:  The authors note: "the overestimation of discrimination may suggest the stomatal slope in the Ball-Berry model (m=9 in Eq. 4) used for these simulations was too high." While it is may be true that the slope parameter is poorly informed by site data, the logic of this conclusion in itself may not be valid. Isotopic measurements *should* give lower slope values than those one would infer via leaf gas exchange data (i.e. the data used to inform the Ball-Berry model). This is because leaf gas exchange measures the resistance from the intercellular spaces (Ci), whereas isotopes measures the resistance from the chloroplast (Cc). I see no mention of this in the text and caution against the authors potential drawing the wrong conclusion from the model-data discrepancy.

Author:  We now address mesophyll conductance specifically as discussed above.  First, our finding that the stomatal slope parameter value is likely too large is a reasonable conclusion for 2 reasons:  1) A lower stomatal slope value is consistent with both model results (Mao et al. 2016) and leaf-gas exchange measurements (Lin et al. 2015).   Discussion of the Lin et al paper was added to page 18 lines: 23-29.  Second, a lower stomatal slope value will lead to a lower stomatal conductance which will help reduce the overestimation of the modeled isotopic discrimination (Figure 8).

With that being said, we have added to the discussion (page 19, lines 3-17) the possibility that this result may, in part, come from the simplified approach of CLM 4.5, that does not specifically include mesophyll conductance and assumes intercellular $CO_2$ = intracellular $CO_2$.   Therefore, we have added the caveat, that the need to reduce the stomatal slope, may be the result of missing mechanisms governing mesophyll conductance within CLM.  We include this possibility in the abstract (page 2, lines 11-14) and conclusions (page 25, lines 28-31).

Referee 2.9:  In discussing the "limited nitrogen formulation", the authors note: "In general, there were no categorical differences in behavior between these two classes of models during CO2 manipulation experiments held at Duke forest and ORNL (Zaehle et al., 2014). CLM 4.0 was one of the few models in that study to consistently underestimate the NPP response to an increase of atmospheric CO2 due to nitrogen limitation, however this finding was attributed to a lower initial supply of nitrogen." This is not strictly true.  As part of the same model-data inter-comparison of the models to the data at the two FACE sites, De Kauwe et al. (2013, Global Change Biology), found no support for the implementation whereby assimilation is limited by nitrogen availability, but not stomatal conductance. They concluded: "Stomatal conductance data from both sites were used to test modelled leaf-level responses. The simple stomatal conductance model (Eq. 1) fitted the data well (Fig. 6), supporting the assumption of coupling between assimilation and stomatal conductance. Importantly, at the ORNL site, N content of the foliage declined strongly over the course of the experiment (Norby et al., 2010), but neither the slope of the stomatal model, nor the response of A/gs to CO2, was altered by this decline (Fig. 6b). These data indicate that the coupling between stomatal conductance and assimilation is not affected by N-limitation (Fig. 6b). The data therefore tend to support coupled models over uncoupled, or partially coupled, models such as DAYCENT and CLM4." Furthermore, I would question if there is any evidence that plants follow the "limited nitrogen formulation"?

Author:  Thank you for bringing De Kauwe et al. 2013 to our attention.  First, we may be talking about two different sub-groupings of models: in our manuscript we are comparing pre-photosynthetic (foliar nitrogen) and post-photosynthetic nitrogen limitation models.  Almost all of these models regardless of pre/post photosynthetic sub-grouping contain stomatal-photosynthetic coupling though Ball-Berry type assumptions in the stomatal conductance model.

It is true that CLM4 and CLM4.5 in the default model (post-photosynthetic model formulation) is only 'partially' coupled in terms of photosynthetic-stomatal conductance, however the unlimited nitrogen formulation (pre-photosynthetic) in our manuscript is 'fully' coupled ($A_n$ is consistent and solved simultaneously with $g_s$).  Therefore, our simulations were consistent with De Kauwe et al. 2013 in that fully-coupled models matched the observations the best.  We add this to the discussion on page 22-23, lines: 30-32, 1-3.

This progressive de-coupling between $A_n$ and $g_s$ for our default CLM 4.5 version also explains the difference in transient behavior between $g_s$ and $A_n$ and iWUE as shown in Figure (7).  We add this to discussion in page: 16, lines: 26-32.

This referee makes another comment that seems to be referring to a 3$^{rd}$ sub-grouping of model – models that do not consider nitrogen limitation at all – similar to the simple stomatal-assimilation model in De Kauwe.  What role does nitrogen limitation play (if any) in assimilation and stomatal behavior?  We are not sure how much our manuscript can inform this question.   Clearly, the default version of CLM 4.5 (post-photosynthetic formulation) is strongly influenced by the nitrogen cycle, whereas our pre-photosynthetic formulation is less strictly linked to the nitrogen cycle (Vcmax was calibrated to match eddy covariance flux observations, not according to nitrogen constraints).   However, even for the pre-photosynthetic formulation it is plausible that leaf nitrogen content plays a role in the Vcmax value. Therefore within the limitations of this manuscript we don't think we can comment on the significance of nitrogen limitation on ecosystem behavior, but only that if nitrogen limitation is implemented, it should occur pre-photosynthetically.

Referee 2.10:  Figure 2. I realize that a strength of this paper is the long time series; however, showing ~15 years of data like that isn't particularly instructive. It is hard to distinguish the model-obs differences. Perhaps average a day/week or monthly climatology across years would more clearly show differences. This figure could also be kept, perhaps one could go to the supplementary.

Author:   As suggested we edited figure 3 to provide a seasonally averaged flux behavior across all years, to better illustrated model-observation differences, and calibrated/uncalibrated model differences.   We moved the original figure 2 within the supplement (Figure S1), not only to provide the length of the data record, but also to demonstrate any transient behavior as revealed by the flux data (i.e. changes in productivity, or latent heat exchange by drier conditions etc.)

Referee 2.11:  Figure 8e. I find it hard to believe that there is no reduction in the soil moisture availability factor during the whole of the summer? This seems unlikely to me? Could this please be checked?

Author:  We thought the same thing. The modeled soil moisture tends to compare favorably to the observed soil moisture as measured at multiple depths both in terms of the magnitude and seasonal trends.   In general, the modeled soil moisture tends to simulate slightly wetter conditions as we point out on page 22, lines 3-12, but per discussion with site PI's, the accuracy of the soil moisture sensors is questionable.   Therefore we did not attempt to calibrate the hydrology model to best match the soil moisture sensors.  We hypothesize that the modeled soil moisture is too wet at depth, thereby leading to little change in BTRAN (soil moisture stress parameter).  We chose to limit this discussion in the text to keep the manuscript length in check, a concern for this reviewer.

Referee 2.12:  Figure 9. I would suggest the symbol sizes could be reduced, they seem a little large for the figure panels.

Author:  Symbol sizes reduced in figure 10.
* * *
Response to comments by Ralph Keeling:

Ralph Keeling 1:  Overall, very nice paper, which I found quite educational. I'm passing on here a few comments that I jotted down as I read through the paper.

Author:  Thank you.

Ralph Keeling 2:   Perhaps it would be possible (?) to add a figure which diagrams the carbon flows from the atmosphere, through stomata to substrate formation for each of the three formulations. I'm imagining that the diagram would have arrows for each of these quantities: An, GPP, CFavailable_alloc,

CF_alloc, CF_GPPpot, etc. Or maybe one figure would suffice, assuming the knobs to switch between formulations is clear enough.

Author: We have added a new figure (Figure 1) which explicitly tracks the carbon flows through the 2 main nitrogen sub-models used in this study, and illustrates how the nitrogen limitation model interacts with these carbon flows from substrate to biomass. We also added new equations (6, 9) which explicitly show the linkage between $A_n$ and $CF_{GPPpot}$ and between $CF_{GPPpot}$ and GPP. This new figure, and the new equations, combined with the existing Table 2 provides a sufficient overview of the sub-models that complements the text to enhance reader understanding. We refer to this new figure on page: 8, line: 13-14 and page: 8, line: 25-26.

Ralph Keeling 3: ….where is the carbon that is fixed but not allocated ending up? Is it respired? If so, does this respiration return back from the stomata or return through some other pathway?

Author: For the limited nitrogen sub-model (post-photosynthetic limitation), the carbon that is fixed but not allocated, is removed from the system (does not show up as a respired flux). This is arguably a weakness in this version of the model: the downscaled assimilated flux is not consistent with the carboxylation rate ($A_n$) and stomatal conductance ($g_s$) that created the pre-downscaled flux, and is why this version of CLM is considered to be 'partially' coupled (page 16, lines 26-32). The unlimited nitrogen sub-model (pre-photosynthetic limitation) is not subject to this apparent inconsistency and is 'fully' coupled. We have added Figure (1) that shows a valve for this downscaling, and no respired flux. We also added a line on page 8, line 24-25 that states that this excess carbon is lost to the system, and does not show up as a respired flux.

Ralph Keeling 4: Page 6, line 28: I'm missing how An is related to terms in Eq. (6). It would very much help to include an algebraic expression for this.

Author: We added new equation (6) that relates An to potential GPP term $CF_{GPPot}$ .

Ralph Keeling 5: Page 6, lines 29-30. From the wording it sounds like maintenance respiration is partly double counted.

Author: The $CF_{GPP,mr}$ term comes directly from the carbon pool from photosynthesis. When there is no photosynthesis the model calls on a storage carbon pool to meet this demand: $CF_{GPP,xs}$. The maintenance respiration is coming from one or the other, never both. We clarify this on page 7, lines 4-6.

Ralph Keeling 6: Page 7, line 11. What does the subscript psn signify? Perhaps could be omitted?

Author: $P_{sn}$ stands for photosynthetic fractionation. This is implied in the context of the manuscript so it was removed throughout.

Ralph Keeling 7: Page 7, line 15: This formula suggests that An is not equal to the flux through stomata. So what is An equal to? Is it the same as potential photosynthesis? If so, needs stating. See earlier comment also.

Author: $A_n$, as defined in equation (1) is the (potential) leaf-level net assimilation rate which is used to calculate potential photosynthesis ($CF_{GPPpot}$). We now specify on page 5, line 21 that $A_n$ is the leaf-level net carbon assimilation.  We also added equation (6) that connects $A_n$ with $CF_{GPPpot}$ based on a previous comment making it clear that $A_n$ is the assimilation rate that is used to calculate potential GPP.

Ralph Keeling 8:  Page 7, lines 22-23: This sentence is a bit ambiguous. Are both given in Eq. 9, or just one. If not both, then how is nitrogen limitation incorporated? Reading below, I see this is probably related to control of Vcmax. If so, this need stating more clearly earlier.

Author:  Both formulations follow Equation (9).  We made changes in the text to make this clearer on page 8, lines 8-10.   "The unlimited nitrogen formulation also follows equation (11), however the vegetation is allowed to have unlimited access to nitrogen."

The following paragraph on page 8 gives a thorough explanation how we used Vcmax in order to take into account for nitrogen limitation, even when the nitrogen downregulation factor is not used.

Ralph Keeling 9: Page 8, line 22. I'm missing an expression for how delta_GPP is calculated from alpha_psn. (Okay, I know enough to work this out for myself, but I'm not sure you should assume all readers would).

Author:  The fractionation factor α is defined in the beginning of section 2.1.2 (page 7, lines 15-16), stating that $\alpha = R_a / R_{GPP}$.  This relates α to $R_{GPP}$. One can then use equation (10) to get from $R_{GPP}$ to $\delta_{GPP}$.

Ralph Keeling 10:  Page 8, line 28. It would seem important to clarify what is meant here by GPP. Which of these is it: An, CFavailable_alloc, CFGPP_pot, etc. ?

Author: This GPP is the final downscaled GPP or actual GPP (ecosystem photosynthesis). We added a new equation (9) that defines how GPP is downscaled through $f_{dreg}$, and this downscaled, or 'actual' GPP is what is used in the definition for WUE.

[revised manuscript text omitted]
_2$ ($c_i$, within chloroplast) is the same as intercellular $CO_2$ (inside leaf stomata, outside chloroplast), but when it can be significantly lower (Di Marco et al., 1990; Sanchez-Rodriguez et al., 1999). The overestimation of $c_i$ could have two important impacts upon our simulation. First, this may lead to unrealistically low values of $Vcmax$ in order to compensate for the overestimation of $c_i$. In fact, we reduced the default value of $Vcmax$ as much as 50% in our simulation to match the eddy covariance flus tower observations (see Section 3.3). Second, the overestimation of $c_i$ should cause an overestimation of discrimination (Eq. 10), consistent with which we have also observed in our simulations (Figure 8). To determine whether the simulated discrimination bias is a model parameter calibration issue ($g_s$) or from a missing representation ofexcluding $g_m$, we recommend additional leaf gas exchange measurements be made at Niwot Ridge to better constrain the stomatal slope value. Furthermore, it would be instructive to include 
[revised manuscript text omitted]

---

## Author Response (AR2)

Response to Associate Editor Comments

An observational constraint on stomatal function in forests: evaluating coupled carbon and water vapor exchange with carbon isotopes in the Community Land Model (CLM 4.5)

Brett Raczka, Henrique F. Duarte, Charles D. Koven, Daniel Ricciuto, Peter E. Thornton, John C. Lin, David R. Bowling

Manuscript #: doi:10.5194/bg-2016-73, submitted Mar 22, 2016

Thank you to the associate editor, Dr. Sonke Zaehle for an exceptionally thorough review.   We have organized the editor's comments with our responses in order from 1 through 37 below.   In our opinion the three most important comments from the editor are, first, a need for a description in the methods of the photosynthetic parameters Aj ($J_{max}$) and Ap.  We address this in editor comments 1, 19 and 29.   Second, the description of our $V_{cmax}$ adjustment procedure as a calibration procedure, and the impact of this calibration upon our results in editor comments 5, 6 and 29.   Third, the description of our unlimited nitrogen formulation within CLM as analogous to a foliar nitrogen model in editor comments 4, 31 and 33.  All pages and line numbers refer to the most recent version of the manuscript, and therefore the page and line numbers referred to by the editor comments are no longer valid.

In addition to the editors concerns, we identified a small error in the previous manuscript draft which required edits to Figure 8, Figure S4 and sections 3.2.2 and 3.2.3 (page 1, line 26; page 19, line 11; page 20, lines 2-4).  We sent an email to the editor on 7/28/16 notifying him of this correction which we paraphrase here:  The modeled photosynthetic discrimination should have been approximately 2 ‰ lower than what was shown in the previous manuscript draft. This does not change the overall discussion or implications within the manuscript --the model simulation still overestimates the observed photosynthetic discrimination.

 Thank you for the opportunity to publish in Biogeosciences.

On behalf of all authors,

Dr. Brett Raczka brett.raczka@utah.edu

University of Utah

Response to general comments of editor:

Editor: Overall I feel that the manuscript has improved. However, I have a list of comments that require addressing before this manuscript can be published in Biogeosciences. While most of these are mostly editorial matters, some require some further attention and clarification.

Author: We thank the editor for his attention to detail and for providing comments that have improved the clarity of the manuscript.

Editor 1: - P6 Why is only Ac explained, but not Aj or Ap? Are they not relevant for the study? I think these should be mentioned, though for the sake of the paper, eq 2 and 3, as well as their Aj and Ap counterparts could become part of an methodological Appendix.

Author 1: We made a judgment call on where to make a cutoff for equations. Our goal was to include those that were most relevant to our analysis. We defined (through text) Aj and Ap (page 5, lines 19-23) , and also provide the reader Oleson et al. (2013) as a source for further details –a manuscript freely available on the CLM website without a journal subscription. We included equation (2) to define $A_c$ (instead of $A_j$ and $A_p$) because 1) this links $V_{cmax}$ to net assimilation ($A_n$) 2) $V_{cmax}$ becomes important later on because of our calibration procedure. In this way we think we achieved a good balance of brevity and clarity relevant to our main focus. In an effort to better address Aj and Ap we add text to page 6, lines 5-6 and page 13, line 12-14 that explains our calibration procedure influences not only $V_{cmax}$ but Aj and Ap as well.

As suggested we could create a methods appendix for equations (2) and (3) but feel inclusion in the main text was necessary given the importance of our calibration procedure, and also feedback from the first line of reviews including from Ralph Keeling that requested some of these equations in the main text.

Editor 2: - Eq 11: Explain why An is assumed to be reduced, but gs not. This violates eq 4.

Author 2: Just to be sure we are on the same page, technically, within Eq. 11 $A_n$ is NOT assumed to be reduced. Both $A_n$ and $g_s$ within eq. 11 directly follow from the coupling in eq. 5. However, and this is a critical point, the coupling between $A_n$-$g_s$ in equation (5) happened BEFORE the nitrogen limitation step in eq. (9). The fundamental reason for why the (1-$f_{dreg}$) is necessary in equation (11) is because this version of CLM only has partial or temporary coupling between An-gs. As described on page 8-9, lines 16-32 and 1-2 and page 16 lines 16-22 and also in previous reviewer/editor responses the stomatal conductance (eq. 5) is solved prior to the nitrogen downregulation step eq. (9). Therefore prior to the nitrogen downregulation the model is still fully coupled between An-gs. However, once the GPP is downregulated through nitrogen limitation, the model is technically un-coupled, because the reduced $A_n$ (call it $A_n$*) is no longer consistent with $g_s$ in equation (5), hence partially-coupled, or as you say eq 5 is violated. In order to account for this within the discrimination part of the model (eq 11) the (1-$f_{dreg}$) term adjusts the $A_n$ (solved for in Equation 5).   In effect the (1-$f_{dreg}$) artificially reduces $A_n$ to $A_n$* therefore artificially boosting ci to ci*.

We apologize if this was already clear and what you may actually be asking is:  why does ONLY $A_n$ have the (1-$f_{dreg}$) coefficient and not $g_s$ in (eq. 11)?  In other words why is $A_n$ adjusted to $A_n$* and $g_s$ not adjusted to $g_s$*?  This is admittedly a potential weakness in the model and we believe one of our key findings.   The allocation downscaling approach to nitrogen limitation, in effect, forces a decoupling between An-gs, and the impact of that decoupling has implications upon the discrimination, which in part is accounted for with the inclusion of the (1-$f_{dreg}$) expression in equation 11.    Based upon our findings in this manuscript the (1-$f_{dreg}$) is not sufficient.  We believe it is worth testing a foliar nitrogen model (which allows for a dynamic nitrogen budget linked to $V_{cmax}$ and full coupling between An-gs) within the framework of CLM (Ghimire et al. 2016) and we say this in pages 22-23 lines 28-31; 1-2 and in the conclusions on page 26, lines 13-16.   We hope to test this another manuscript, but it is out of the scope for this manuscript.

Editor 3:  Also verify scaling coefficients for gb and gs. My textbook says the diffusion correction factors are 1.37 for boundary layer and 1.6 for stomate's whereas your equation suggests the inverse. This will obviously have a fairly significant effect on the predicted ci and thus discrimination!

Author 3:  We double-checked and this is how the coefficients are defined in equation (25.10) of Oleson et al. (2013), and also the way they are implemented within the source code of CLM 4.5. Equation (25.10) comes directly from equation (8.31) in Oleson by solving for ci.  The terms 1.4 and 1.6 are the ratios of diffusivity of $CO_2$ to $H_2O$ for the leaf boundary layer resistance and stomatal resistance respectively,  however because equation (25.10) is expressed in terms of conductances (1/resistance) the coefficients are reversed and therefore 1.4 is multiplied by the leaf stomatal conductance and 1.6 is multiplied by the leaf boundary layer conductance.

Editor 4: - P8 L10: It is unclear where this nitrogen is then coming from? Does this not break the logic of CLM 4.5's closed N budget and introduce an unknown source of N? Or is this only assumed in eq 11 (then be more precise as to how this is implemented, and what happens to the extra C assimiation).

Author 4:    It is important to distinguish between the two different model formulations, the *limited nitrogen* formulation and the *unlimited nitrogen* formulation, which are two entirely separate CLM simulations.   As we have described on page 8-9 lines 16-32, 1-2 and also illustrated in Figure 1 the *limited nitrogen* formulation calculates a potential amount of allocated carbon (CF_avail_alloc, Eq. 8 ) and then the actual amount of allocated carbon ( CF_avail, Eq. 8) based upon nitrogen availability.   This 'extra' carbon as you call it (i.e. the difference between $GPP_{pot}$ -GPP) is instantaneously 'lost' to the system (i.e. does not show up as a respiration flux etc.)  We state this on page 8, line 27-29.

The unlimited nitrogen formulation ignores the nitrogen budget entirely and assumes that there is an unlimited amount of nitrogen in the system (i.e. nitrogen does not impose a limit on the allocation at all). We described this in page 8 lines 8-15. CLM 4.5's closed N budget is not considered in this case, and your intuition is correct to ask: when this 'extra' nitrogen is provided are the results realistic? The answer is no. If provided unlimited nitrogen CLM is much too productive (See Figure S2; uncalibrated run for unlimited nitrogen formulation). That is why we performed a separate $V_{cmax}$ calibration for the *unlimited nitrogen* simulation to provide reasonable productivity (see page 13, lines 3-10). In this way although the *unlimited nitrogen* formulation ignores CLM 4.5's closed N budget, it still parameterizes the effect of nitrogen limitation within the $V_{cmax}$ calibration. This is more similar to a foliar nitrogen style model as we discuss on page 22, lines 28-31, 1-2. We have added text to clarify that although the *unlimited nitrogen* formulation ignores CLM 4.5's nitrogen budget, it still accounts for nitrogen limitation through the $V_{cmax}$ parameterization (page 8 line 8-12).

We recognize this is a difficult point to grasp, but hopefully the above explanation and manuscript edits have made it clearer.

Editor 5: A further point needing attention is that I think labelling your $V_{cmax}$ tuning a calibration is misleading, because the calibration directly includes the observed and simulated GPP as a factor, depends on time-varying site-data, and is not useful outside without site-data. This approach does override a lot of the model's internal dynamics and exceeds what is conventionally called "calibration". It is not surprising that the "calibrated" model performs better, because you prescribe the seasonal course of GPP. Section heading 2.4 and associated text is hence misleading.

Author 5: We agree with the editor that the $V_{cmax}$ calibration approach is limited in its applicability given it was created based on simulated/observed GPP at Niwot Ridge. We state this in the manuscript on page:14 lines: 1-4 '.....***within the confines of our study area***, our calibration approach was sufficient to provide a skillful representation of photosynthesis and provided a sufficient testbed for evaluating carbon isotope behavior.' To further clarify this point we have added an additional line cautioning against the application of this calibration outside of the study area. Page: 14, lines 3-4.

Although our calibration approach may not be as common, it is a time-varying parameter optimization approach that falls under the general umbrella of calibration where the end goal is to use observations to adjust parameters to reduce model-observation mismatch. Just like any other calibration approach it can correct for inaccurate parameters, and sometimes (both directly/indirectly) for structural errors within the model. For example, the approach to force the winter-time GPP to zero, in effect compensating for structural errors, is a more aggressive approach, however has been done before (Kolari et al. 2007). Therefore we suggest we keep the calibration title in 2.4 as this is well recognized within the community, and add the new text that emphasizes the limitation in our approach.

Editor 6: Because this is the case, I don't think that there is a need to emphasize differences between the off-the-shelf CLM 4.5 and your improved version. This is particularly the case as this manuscript focusses on the use of 13C as a constraint. I therefore don't see the relevance of presenting the off-the-shelf parameterisation in 3.1. specifically if the model behind the seasonal dynamics of GPP are fundamentally different between the two. The discussion of how to improve the seasonal representation of GPP also seems off topic as it does not involve the use of isotopes, and I would therefore suggest to drop this.

Author 6:  We agree, and eliminated detailed discussion (~8 lines) that compared the difference between the calibrated and uncalibrated model runs in an effort to focus on isotope results.   We also cut approximately 6 lines of discussion concerning the seasonal representation of GPP in an attempt to stay on topic.   We kept the discussion about how our calibration approach could be improved by addressing the underlying structural problems within the model.   This is useful because it emphasizes the limitations of our calibration approach (as the editor pointed out earlier) and also provides other modelers with insights on how to improve model performance.

Editor 7: The results section contains large stretches of classical discussion (Sections 3.3 and 3.4). They should be part of a discussion section and not the results, as such as no new evidence is presented and they distract the reader from the results you present.

Author 7:  We moved Sections 3.3 and 3.4, previously a part of the Results & Discussion section, to the Discussion section.  They are now Section 4.1 and 4.2.

Editor 8:  The abstract is currently very long. Please try shortening it a bit. Pre- and post-photosynthetic N limitation are CLM construct and thus not clear to the general readership. I would reformulate this to read that the default $A_n$-$g_s$ coupling of CLM failed, but a revised formulation coupling $A_n$ and $g_s$ did successfully capture the observed dynamics without mentioning the CLM-terms, which are ok to use (though still awkward) in the manuscript).

Author 8:   We reduced the abstract by removing 2 non-essential sentences concerning soil moisture and WUE/discrimination trends.  We also included the coupling terminology suggested by the editor. In addition we have explicitly stated in the Methods on page 8-9 lines 29-32, 1-2 that the limited nitrogen formulation has partial coupling between the carbon-water cycle and that,  the unlimited nitrogen formulation is fully-coupled.  Before it would not have been clear until the discussion that this was the case.   We also use the partial/fully coupled terminology in the conclusions for consistency (page: 25  line: 17-19),  and add $A_n$-$g_s$ coupling as a column in Table 2 where we thought this was more descriptive than the previous column heading in Table 2 which was 'Impact upon $A_n$-$g_s$'.
* * *
* * *
Response to detailed comments of editor:

Editor 9: P3 L 11 (add "depending on for instance" to both brackets).

Author 9: This change was made.

Editor 10: P3 L 17 Provide a reference for the fact that discrimination is indeed affected by nutrient availability in the real world.

Author 10: We added Cernusak et al. (2013).

Editor 11: P4L7ff can be removed.

Author 11:   We removed this line.

Editor 12: P4 L20: It does not really become clear here why you would think this is an issue that needs particular addressing. Isn't is logical that the offset between the 13CO2 at site and global mean values must be important when looking at the absolute delta13CO2 values? So why would one use a global mean value to start with?

Author 12:   We intended to mean that given site specific boundary conditions (including for instance site-level atmospheric $\delta^{13}CO_2$) and the stable carbon isotope sub-model within CLM, could the observed $\delta^{13}CO_2$ of carbon pools be simulated.  We clarified this sentence in the text. (page: 4 line: 17-21)

Editor 13: P4 L24: water-use efficiency

Author 13:  Changes were made here and elsewhere in text.

Editor 14: P6 L 14ff: Include beta_t into eq 3 if it's used there. Otherwise the current paper reads as if only the intercept of the Ball&Berry relationship is affected by soil moisture. While doing this, clarify whether beta_t also affects Ap and Aj?

Author 14:    Yes, beta_t is applied to Vcmax, but is applied much later in the CLM source code, hence we left it out of eq. 3 originally.  We include it now.  Beta_t only influences $V_{cmax}$ and Ac, not Ap and Aj within CLM.

Editor 15:  P6: Equation 6 define LAIsunlit/shade how calculated or provide a reference.

Author 15:  We added Oleson et al. (2013) as reference.

Editor 16:  P7 L22: Provide a reference for these values. Explain how ci depends on shaded and sunlit leaves, and how this affects the leaf ci calculations and total discrimination of the An flux?

Author 16: We add the Farquhar et al. (1989) reference for the diffusional and enzymatic values. We add text (page: 8 lines: 1-4 ) describing that the $c_i$ is calculated for shaded and sunlit portion of the leaves separately, therefore discrimination is calculated separately. What we show in the equation (11) is a simplified, general expression for simplicity and brevity.

Editor 17: P8 L7: should this not read nitrogen-limited rather than limited nitrogen? Assuming that the actual N-limitation of the ecosystem is identical between the runs, and its only a question of whether GPP is assumed to be down-regulated with N limitation or not? It seem so because that's what is said later?

Author 17: We fail to see the difference between nitrogen-limited vs. limited nitrogen. The N-limitation is in fact, not identical between the two runs. The limited nitrogen simulation downregulates GPP by tracking nitrogen within CLM, whereas the unlimited nitrogen has no downregulation of GPP, but instead depends upon $V_{cmax}$ calibration. We discuss this in detail in page 8 lines 5-32.

Editor 18: P8 L13: Add a note that all this is very specific to the set-up of CLM and not representative of the wider class of biosphere models.

Author 18: The unlimited nitrogen formulation of CLM is specific to this analysis only, whereas the limited nitrogen formulation is the default version of CLM. We state this specifically now in the text to reinforce this point (page: 8 line: 8-15).

Editor 19: P12 L28: What happened to Jmax?

Author 19: We added text (page: 13, lines: 12-14) stating that both Aj (and $J_{max}$) and Ap defined in equation 1, are also calibrated in the same way as $V_{cmax}$ described in Section 2.4.

Editor 20: P13 L14-22. Not necessary

Author 20: This section was removed.

Editor 21: P15 L31. The FLUXNET data do not provide evidence for the response of stomates to CO2, they simply suggest a trend in WUE with time, which may (or may not) be related to increasing CO2. I would rather recommend to quote a study relating stomatal conductance to elevated CO2 experiments.

Author 21: We replaced the Keenan et al (2013) citation with Ward et al (2013), a Duke FACE site experiment.

Editor 22: P16 L 1 Is this change in ci:ca really a consequence of changes in CO2, or concurrent climatic changes? Because eq 4 should imply constant ci:ca...

Author 22: Yes, because on a multi-decadal time scale the meteorology we used in the model was constant –that is we repeatedly cycled the site-level meteorology between 1998-2013 for the time period of 1850-2013.  We state this in the methods (page: 11  lines: 24-29)  and also within the discussion in the preceding section (page: 15 lines 15-19) and afterwards (page: 16, 2-5).

Editor 23: P17 L 7: In the model, or in the data? Explain briefly how VPD is affected by your model formulation - not everybody will get this.

Author 23:  We intended to mean in general, both in the real world and for the model.  We add in parentheses how VPD influences the stomatal conductance within the model (page: 16-17 lines: 28-31, 1-2).

Editor 24:  P17 L12: This is a good example of a paragraph that would be better suited in a different, separate discussion section, as it is distracting from the line of results presented. This is particularly true because another discussion of gm is on P19 L3ff

Author 24:  As suggested earlier we have created a designated Discussion (Section 4) that is separate from the Results & Discussion (Section 3) which should help the organization and focus overall.   In the case of mesophyll conductance as the editor mentions above, we agree that if one is interested in the results all at once, or if one would prefer the discussion of mesophyll conductance to be combined in one section, then a designated discussion section fore mesophyll conductance would be best.  However, if instead, one prefers fully developed, coherent sections based on concepts such as in Section 3.2.1 where we discuss trends in photosynthetic discrimination, and Section 3.2.2 where we discuss the magnitude/bias in photosynthetic discrimination, then our existing organization would be best.   One might find it distracting to be presented trends and biases in photosynthetic discrimination, then have to scan through the rest of the results/discussion before returning to the concept of mesophyll conductance again.   It is not clear to us one method of organization is more advantageous than the other, therefore, at the editor's discretion, we prefer to keep it as is.

Editor 25:  P17 L30: I must have missed this: which global simulation. So far, you only talked about Niwot ridge.

Author 25:  We introduce the global CESM simulation at the end of the previous paragraph: 'These trends in iWUE and discrimination have also been found in a fully-coupled, isotope enabled, global CESM simulation (Figure S3).'  (Page 17, Lines 12-14)

Editor 26: P18 L 3. Why is this equation introduced here and not in the Methods?

Author 26: We felt it was logical to include this equation only after we had fully developed the discussion between discrimination and iWUE.   Otherwise, if introduced in the methods, it would be premature.  In addition it also may present some confusion to the reader to put the equation in the Methods, given that it is not an equation found in CLM, but a simplification of a CLM equation used as a diagnostic for model behavior.   We also felt the manuscript would be easier to read if we introduced this equation as the clincher to this discussion section, rather than have the reader have to jump way back to the Methods section.

Editor 27: P19 L18. which mixing model?

Author 27: We mean the mixing model approach employed by Bowling et al. (2014) which we introduce at the beginning of this section (page 18: line 3-4). We add the citation again here, to remind the reader where this is coming from (page 19: lines 7-8).

Editor 28: P19 L30ff This may either be the introduction to this section or removed, because at this stage, you are simply repeating what's been said before.

Author 28: Yes, it was our intent to summarize the key points given the complexity of the discussion. From personal experience we have found a condensed summary helps reinforce the points, and at the editor's discretion we prefer to keep this.

Editor 29: P20 L6 ff: Isn't this all simply a consequence of the model tuning via a time-dependent Vcmax while not touching leaf respiration? This must yield unrealistically high ci:ca and thus discrimination values. Given that this is an artefact of the way you've matched the seasonal cycle and it will be specific to CLM 4.5, is it really worth discussing this here?

Author 29: Not only is $V_{cmax}$ ($A_c$) adjusted in our time dependent calibration approach, but also for Aj, Ap and the maintenance leaf respiration. We have added text clarifying this point (page 13: lines 12-14). Both the limited nitrogen and unlimited nitrogen formulations undergo similar $V_{cmax}$ calibrations, therefore the difference in seasonal discrimination performance between the two formulations being an artifact of the calibration seems unlikely. Instead, as we state in the manuscript we believe the difference in behavior is likely the result of the nitrogen downregulation approach used in the default model, and the method for accounting for the downregulation approach within the ci equation (eq. 11). Furthermore, the fact that the nitrogen downregulation also forces the model to be partially-coupled where $g_s$ is no longer consistent with $A_n$ may also play a role.

Editor 30: P22 L25. Please define what you mean by "categorical differences." It is not correct to state that CLM4 did not capture the NPP response only because it had too little initial N availability. We have demonstrated clearly that it was lacking important functional features of the response such as the too strong carbon-nitrogen coupling of biomass production (Zaehle et al. 2014). CLM4 had the lowest WUE response in the ensemble (de Kauwe et al, 2013).

Author 30: Thank you for the correction. We have changed the text to say that the poor NPP response was from both initial N availability and too strong carbon-nitrogen coupling. (page 22: line: 20-21)

Editor 31: P23 L4. This is not correct, because you do not update foliar N to calculate Vmax (with all the consequences for the N requirement of biomass etc.), but prescribed a time-dependent Vcmax scaled to the observed GPP independent of foliar N (unless your description in Sect 2.4 is inaccurate).

Author 31:  We have changed the wording to say there are only 'similarities' between the unlimited nitrogen formulation and a true foliar nitrogen model.   We then point out that a true foliar nitrogen model would link a dynamic nitrogen cycle with leaf nitrogen (Ghimire et al. 2016) (page: 22-23 line: 28-31, 1-2)

Editor 32:  P25 L22: No land surface model qualifies as highly mechanistic.

Author 32:  We removed the word 'highly'.

Editor 33:  P25 L22L What you have shown is that if employing eq 4 on the "true" An, all is fine, but if you follow the standard CLM approach, the results are not as good. What you have not shown is that if one uses foliar N to diagnose $V_{cmax}$, this improves the simulation of discrimination. You have not, because you calibrate for each day $V_{cmax}$ based on its difference to the observed GPP, implying that there are variations of $Vcmax,25$ on a day-to-day basis unrelated to any changes in foliar N. The latter would be what "foliar-N" models do. Therefore, your claim that you are mimicking the functionality of a foliar N model is unsubstantiated.

Author 33:  We agree that the unlimited nitrogen formulation of CLM and a foliar nitrogen model are not the same thing, although there are similarities.   We remove the text 'mimic the functionality of a foliar nitrogen model' to clarify this. (page 25, line 14-16).

Editor 34:  Figures: Please label all sub-plots with consecutive letters and refer to these in the caption.

Author 34:  We made these changes for all figures.

Editor 35:  Figure caption 1: "The N-limitation is determined by required N availability to meet demand from C:N ratio based on CF." Isn't the available soil N determining whether or not the growth is N limited?

Author 35:  We agree that what you have suggested is a better wording.  We have updated the caption in Figure 1 to say 'N-limitation is applied if the available N cannot meet the demand determined by the available carbon for allocation () and the C:N biomass ratio'.

Editor 36: I wonder if this Figure could be made clearer by having two arrows between the box of eq 8 and eq2 indicating for both scheme the direction if information (ie is it growth or Vcmax determining photosynthesis and conductance?).

Author 36: Not sure if this is exactly what you mean, but we like the idea of a backwards arrow linking the downregulation nitrogen box to the photosynthesis-stomatal conductance box (feed-back loop), to indicate that photosynthesis is reduced.  We have made this change.

Editor 37:  General remark: carefully revise writing and punctuation

Author 37: We have carefully revised the entire document.  Thank You.

[revised manuscript text omitted]

---

## Author Response (AR3)

Response to Associate Editor Comments

An observational constraint on stomatal function in forests: evaluating coupled carbon and water vapor exchange with carbon isotopes in the Community Land Model (CLM 4.5)

Brett Raczka, Henrique F. Duarte, Charles D. Koven, Daniel Ricciuto, Peter E. Thornton, John C. Lin, David R. Bowling

Manuscript #: doi:10.5194/bg-2016-73, submitted Mar 22, 2016

Our response and associated manuscript changes for the two minor editor comments are described below.   We are very pleased with how the manuscript has improved since our first submission.   We thank all the reviewers and the editor for their efforts.

 Thank you for the opportunity to publish in Biogeosciences.

On behalf of all authors,

Dr. Brett Raczka brett.raczka@utah.edu

University of Utah

Response to editor comments:

Editor 1:  point #2 [from previous comments]: The text explaining eq 11 still need to be modified to state that eq 11 attempts to mimic eq5, but fails to do so. Much of what you are investigating is the unintended consequence of a model implementation, which focusses on N limitation for biogeochemical, but not biogeophysical processes. It needs to be clear to the reader that one could have derived a "correct" implementation by adjusting gs in concert with An, and therefore that eq 11 does not satisfy eq 5.

Author 1:  We have added to the text that the mismatch between the actual photosynthesis and the intracellular CO2: …" is a result of the carbon water (An-gs) coupling (eq. 5) being imposed prior to the effect of nitrogen limitation (eq. 9) and is an artifact of the model implementation. We also test a separate model formulation (described in detail in the next paragraph) specific to this analysis that imposes nitrogen limitation through the $V_{cmax}$ parameterization and removes the artifact of $f_{dreg}$. " (Page 8. Lines 1-8).

Editor 2:  point #25 [from previous comments]: The boundary conditions for the global simulation are not clear. If there is no published reference, then in the main text, include,

[…driven by observed historical climate etc.… , (unpublished, K. Lindsay 20XX)]. Because the global simulation was not introduced in the Methods, I would prefer if you could also state "a global CLM4.5 run with historic climate" the second time you mention this.

Author 2: We have made this change. (page 17, line 13-16, page 17, lines 24-26). We now specify that the climate is driven by $CO_2$ emissions, and simulated as an emergent property of CAM5 –the atmospheric model within CESM.

Thank You.

[revised manuscript text omitted]